# Implementation of the ISORROPIA-lite Aerosol Thermodynamics Model into the EMAC Chemistry Climate Model (based on MESSy v2.55): Implications for Aerosol Composition and Acidity

Alexandros Milousis[1], Alexandra P. Tsimpidi[1], Holger Tost[2], Spyros N. Pandis[3,4], Athanasios Nenes[3,5], Astrid Kiendler-Scharr[1+], and Vlassis A. Karydis[1]

[1]Forschungszentrum Jülich GmbH, Institute for Energy and Climate Research, IEK-8 Troposphere, Jülich, Germany
[2]Johannes Gutenberg University Mainz, Institute of Atmospheric Physics, Mainz, Germany
[3] FORTH ICE HT, Institute of Chemical Engineering Sciences, Patras 26504, Greece
[4]University of Patras, Department of Chemical Engineering, Patras 26500, Greece
[5]Ecole Polytechnique Fed Lausanne, School of Architecture Civil & Environmental Engineering Lab, Atmospheric Processes & Their Impacts, CH-1015 Lausanne, Switzerland
[+]deceased

*Correspondence to*: Vlassis Karydis (v.karydis@fz-juelich.de)

**Abstract.** This study explores the differences in performance and results by various versions of the ISORROPIA thermodynamic module implemented within the global atmospheric chemistry model EMAC. Three different versions of the module were used, ISORROPIA II v1, ISORROPIA II v2.3, and ISORROPIA-lite. First, ISORROPIA II v2.3 replaced ISORROPIA II v1 in EMAC to improve pH predictions close to neutral conditions. The newly developed ISORROPIA-lite has been added to EMAC alongside ISORROPIA II v2.3. ISORROPIA-lite is more computationally efficient and assumes that atmospheric aerosols exist always as supersaturated aqueous (metastable) solutions while ISORROPIA II includes the option to allow the formation of solid salts at low RH conditions (stable state). The predictions of EMAC by employing all three aerosol thermodynamic models were compared to each other and evaluated against surface measurements from three regional observational networks (IMPROVE, EMEP, EANET) in the polluted Northern Hemisphere. The differences between ISORROPIA II v2.3 and ISORROPIA-lite were minimal in all comparisons with the normalized mean absolute difference for the concentrations of all major aerosol components being less than 11 % even when different phase state assumptions were used. The most notable differences were lower aerosol concentrations predicted by ISORROPIA-lite in regions with relative humidity in the range of 20% to 60% compared to the predictions of ISORROPIA II v2.3 in stable mode. The comparison against observations yielded satisfactory agreement especially over the US and Europe, but higher deviations over East Asia, where the overprediction of EMAC for nitrate was as high as 4 µg m$^{-3}$ (~ 20%). The mean annual aerosol pH predicted by ISORROPIA-lite was on average less than a unit lower than ISORROPIA II v2.3 in stable mode, mainly for coarse mode aerosols over Middle East. The use of ISORROPIA-lite accelerated EMAC by nearly 5 % compared to the use of ISORROPIA II v2.3 even if the aerosol thermodynamic calculations consume a relatively small fraction of the EMAC computational time. ISORROPIA-lite can therefore be a reliable and computationally efficient alternative to the previous thermodynamic module in EMAC.

**Keywords:** atmospheric aerosols, aerosol thermodynamics, nitrate, acidity, aerosol phase state.

## 1. Introduction

Aerosols in the atmosphere have a significant impact on climate and air pollution. They contribute to the deterioration of air quality, especially in heavily industrialised regions, leading to increased mortality rates and decreased life expectancy (Héroux et al., 2015). Particulate matter with diameter less than 2.5 μm ($PM_{2.5}$) is the largest contributor to stroke, cancer, heart conditions and chronic obstructive pulmonary diseases (Brook et al., 2010; Pope et al., 2011) with ambient pollution causing approximately 4.2 million premature deaths in 2019 alone (WHO, 2022). Tarin-Carrasco et al. (2021) predicted that mortality rates in Europe due to air pollution could increase in the next thirty years in the more extreme emission scenarios (e.g., RCP8.5). In addition to the direct threat aerosols pose to humans and ecosystems through their effects on air quality, they can also affect other climate-related processes. For example, they can act as cloud condensation nuclei and modify cloud lifetime and optical properties (Andreae et al., 2005; Klingmüller et al., 2020). Aerosols also affect the energy balance of our planet by reflecting additional solar radiation back to space and thus cooling the atmosphere or by absorbing solar radiation warming the atmosphere (Klingmüller et al., 2019; Miinalainen et al., 2021). Some major inorganic aerosol components also affect various ecosystems. For example, nitrates and sulfates can harm flora by lessening its lifetime and variety (Honour et al., 2009; Manisalidis et al., 2020), and can affect wildlife by causing water eutrophication (Doney et al., 2007). A critical property of atmospheric particles that regulates their impacts on clouds and ecosystems is their acidity (Karydis et al., 2021). Depending on its levels, acidity can affect air quality and human health (Lelieveld et al., 2015) but also the aerosols' hygroscopic characteristics (Karydis et al., 2016). The aerosol pH also drives the partitioning of semi-volatile inorganic components between the gas and aerosol phases (Nenes et al., 2020). Finally, aerosol acidity plays a role in the activation of halogens in aerosols (Saiz-Lopez and von Glasow, 2012), their toxicity (Fang et al., 2017) and also in secondary organic aerosol formation (Marais et al., 2016).

Sulfate is the most important component of $PM_{2.5}$ inorganic aerosol, since it contributes the most in terms of global mass burden (Szopa et al., 2021) and aerosol optical depth (AOD) (Myhre et al., 2013). Nitrate contribution to the $PM_{2.5}$ aerosol composition is also important in several areas (e.g., Europe, North America, East Asia) and seasons (He et al., 2001; Silva et al., 2007; Weagle et al., 2018; Tang et al., 2021). The quantification of nitrate partitioning between the gas and particulate phases is challenging partly because it is affected by meteorology (temperature, relative humidity) and all ionic aerosol components, but also due to the lack of observations to constrain the composition of the gas-phase components and the size-distribution of the particulate phase. Nitrate in the form of ammonium nitrate is mainly found in the fine mode (e.g., $PM_{2.5}$) (Putaud et al., 2010). This is especially the case over polluted regions where there is enough ammonia remaining after the neutralization of sulfate (Karydis et al., 2011; Karydis et al., 2016). In coastal and desert areas, nitrate is formed mainly by reactions of $HNO_3$ with sea salt and dust particles (Savoie and Prospero, 1982; Wolff, 1984; Karydis et al., 2016) and therefore is found mainly in the coarse particles. The importance of nitrate in the troposphere is expected to increase in the following decades because $SO_2$ emissions are anticipated to drop while $NH_3$ emissions to increase (Fu et al., 2017; Chen et al., 2019; Xu et al., 2020). With decreased $SO_2$ concentrations,

less ammonia is required to neutralize the sulfates and therefore more is available for ammonium
nitrate formation (Tsimpidi et al., 2007).
There have been several thermodynamic models developed in the last decades to calculate
the inorganic aerosol concentrations and composition in the atmosphere. Two of the first were
EQUIL and KEQUIL developed by Bassett and Seinfeld (1983). Then the MARS model was
developed by Saxena et al. (1986) with the aim of reducing the computational time required in
order to be incorporated into larger scale chemical transport models. MARS was the first model to
divide the composition domain into smaller sub-domains aiming to reduce the number of equations
needed to be solved. Then the SEQUILIB model by Pilinis and Seinfeld (1987) was the first to
incorporate sodium and chloride and the corresponding salts in the simulated aerosol system.
Further developments included EQUISOLV by Jacobson et al. (1996) as well as SCAPE by Kim
et al. (1993), which simulated temperature dependent deliquescence following Wexler and
Seinfeld (1991) and predicted the presence of liquid phase aerosols even at low relative humidity
(RH). E-AIM is another benchmark thermodynamic model which instead of solving algebraic
equations for equilibrium, uses the minimization of the Gibbs Free Energy approach (Wexler and
Clegg, 2002). Later versions of E-AIM also include selected organic aerosol components (Clegg
et al., 2003). Furthermore, AIOMFAC is a model that utilizes organic-inorganic interactions in
aqueous solutions in order to calculate activity coefficients up to high ionic strengths (Zuend et
al., 2008) and is based on the LIFAC model by Yan et al. (1999). Further developments in
AIOMFAC include a wider variety of organic compounds (Zuend et al., 2011). The EQSAM
thermodynamic model was developed by Metzger et al. (2002) with the basic concept that aerosol
activities in equilibrium are controlled by RH, and solute activity is a function of RH. The model
uses a domain structure based on sulphate availability to increase computational efficiency by
solving fewer thermodynamic equations, similar to Nenes et al. (1998). EQSAM and ISORROPIA
are the two available options for aerosol thermodynamics in the EMAC model.
Nenes et al. (1998) developed the ISORROPIA model in an effort to increase
computational efficiency while maintaining the accuracy of the calculations. The system simulated
by ISORROPIA included $NH_4^+$, $Na^+$, $Cl^-$, $NO_3^-$, $SO_4^{2-}$ and $H_2O$. ISORROPIA also contains the
temperature dependent equations for deliquescence by Wexler and Seinfeld (1991) and is
computationally efficient so that it can be incorporated in 3D atmospheric models. In ISORROPIA,
the aerosol state is predicted as a weighted mean value of the dry and wet states. The weighting
factors depend on ambient RH, the mutual deliquescence relative humidity (MDRH) and the
deliquescence relative humidity (DRH) of the most hygroscopic salt in the mixture. An improved
version of ISORROPIA including the mineral ions $K^+$, $Ca^{2+}$, and $Mg^+$, called ISORROPIA II, was
developed by Fountoukis and Nenes (2007). The addition of the above crustal ions resulted in the
inclusion of 10 more salts and 3 more ions in the solid and aqueous phases respectively. The model
gained in computational efficiency by performing different calculations for different atmospheric
chemical composition regimes, which are determined by the abundance of each aerosol precursor
as well as the ambient temperature and relative humidity. Depending on the values of the so-called
'sulfate ratio', the 'crustal species and sodium ratio' and the 'crustal species' ratio, five aerosol
composition regimes are determined in order to calculate the necessary equilibrium equations for
the species present in each regime. Furthermore, the use of pre-calculated look-up tables for the
activity coefficients (see Section 2.2), including their temperature dependence, is another factor

for the gain in computational efficiency. Like E-AIM, ISORROPIA II can solve the thermodynamic equilibrium problem under stable or metastable conditions. In the second case aerosols are assumed to exist only as supersaturated aqueous solutions even at low RH, while in the first the aerosols are able to form solid salts. A very slightly updated version, called ISORROPIA II v2.3 was introduced to improve aerosol pH predictions close to neutral conditions (Song et al., 2018). More specifically, in some subcases of the ISORROPIA II regime, $NH_3$ evaporation was not taken into account in the aerosol pH calculations, leading to unrealistic estimates close to neutrality (pH~7). This error had a minimal effect on the predicted gas phase $NH_3$ levels and consequently on the inorganic aerosol concentrations. Moreover, it only affected a few subcases and only when the stable mode was used. More details on these differences can be found in Song et al. (2018). The newest development of ISORROPIA II, called ISORROPIA-lite, was designed to be even more computationally efficient than its predecessor and to also include the effects that organic aerosol components have on particle water and the semi volatile inorganic aerosol species partitioning (Kakavas et al., 2022).

This study aims to evaluate the newly developed ISORROPIA-lite thermodynamic module within the EMAC global climate and chemistry model and to explore any discrepancies on a global scale, by utilizing different aerosol phase states. For this reason, our analysis explores the differences in the results between ISORROPIA-lite and ISORROPIA II over diverse conditions and environments. In Section 2 the model configuration and the treatment of inorganic aerosols thermodynamics is presented. In Sections 3 and 4 the results and comparisons between the simulations are analyzed and in Section 5 the major conclusions are presented.

## 2. Model Configuration

### 2.1 EMAC model setup

The EMAC (ECHAM5/MESSy) model is a global atmospheric chemistry and climate model (Jockel et al., 2006). It includes a series of submodels and links them via the Modular Earth Submodel System (Jöckel et al., 2005) to the base model (core) that is the 5th generation European Center Hamburg general circulation model (Roeckner et al., 2006). Gas-phase chemistry is simulated by MECCA (Sander et al., 2019) with a simplified scheme similar to the one used in CCMI (Chemistry-Climate Model Initiative) like in Jockel et al. (2016). Aerosol microphysics along with gas/aerosol partitioning are treated by GMXe in which the aerosols are differentiated between soluble and insoluble modes with a total of seven lognormal modes (Pringle et al., 2010). The soluble mode contains the nucleation, Aitken, accumulation, and coarse size ranges while the insoluble mode lacks only the nucleation size range. Transfer of material between the insoluble and soluble modes is calculated in two processes. After coagulation, when a hydrophobic and a hydrophilic particle coagulate, the resulting mass is assumed to reside in the hydrophilic mode and also when soluble material condenses onto a hydrophobic particle (after gas/aerosol partitioning) it is again transferred to the hydrophilic mode (Pringle et al., 2010). Wet deposition of gases and aerosols is described by SCAV (Tost et al., 2006; 2007), dry deposition via DRYDEP (Kerkweg et al., 2006) and gravitational sedimentation of aerosols by SEDI (Kerkweg et al., 2006). Cloud

properties and microphysics are calculated by the CLOUD submodel (Roeckner et al., 2006) utilizing the detailed two-moment liquid and ice-cloud microphysical scheme of Lohmann and Ferrachat (2010) and considering a physically based treatment of the processes of liquid (Karydis et al., 2017) and ice crystals (Bacer et al., 2018) activation. The organic aerosol composition and evolution in the atmosphere is calculated by the ORACLE submodel (Tsimpidi et al., 2014; 2018).

The model simulations in this work were nudged towards actual meteorology using ERA05 data (Hersbach et al., 2020). For the purposes of this study the spectral resolution applied within EMAC was the T63L31 which corresponds to a grid resolution of 1.875° x 1.875°, covering vertical altitudes up to 25 km with a total of 31 layers. The simulations were all done for the period 2009-2010, with 2009 representing the model spin-up period.

Anthropogenic emissions of aerosols and aerosol precursors were based on the EDGARv4.3.2 inventory (Crippa et al., 2018). Open biomass burning emissions were derived by the GFEDv3.1 database (van der Werf et al., 2010), and natural emissions of $NH_3$ (volatilization from soils and oceans) were based on the GEIA database (Bouwman et al., 1997). $SO_2$ emissions by volcanic eruptions are based on the AEROCOM dataset (Dentener et al., 2006), as are emissions of sea spray aerosols using the chemical composition proposed by Seinfeld and Pandis (2016). Biogenic emissions of NO from soils are calculated online according to the algorithm of Yienger and Levy (1995) while $NO_x$ produced by lightning is also calculated online based on the parameterization of Grewe et al. (2001). Oceanic emissions of DMS are calculated online by the AIRSEA submodel (Pozzer et al., 2006). Finally, the dust emission fluxes are calculated online according to Astitha et al. (2012), by taking into account the meteorological information for each grid cell (i.e., temperature and relative humidity) as well as the different thresholds of friction velocities above which suspension of dust particles takes place. The emissions of crustal ions ($Ca^{2+}$, $Mg^+$, $K^+$ and $Na^+$) are estimated as a fraction of the total dust flux based on the soil chemical composition of each individual grid cell (Karydis et al., 2016; Klingmüller et al., 2018). These ions are emitted in the insoluble accumulation and coarse size modes and are subsequently transferred to the soluble aerosols by the processes described above.

## 2.2 Inorganic aerosol thermodynamics treatment

In this study, the ISORROPIA-lite aerosol thermodynamic model has been implemented into the EMAC as part of the GMXe submodel, not as a replacement but as an alternative to the previous version, in order to efficiently calculate the equilibrium partitioning of the inorganic species between gas and aerosol phases. Furthermore, ISORROPIA II v2.3 is used to replace ISORROPIA II v1 in the model.

Kinetic limitations in the partitioning need to be taken into consideration because only fine aerosols are able to achieve equilibrium within the time frame of one model time step, which in this study equals to 10 minutes. Therefore, the partitioning calculation is done in two stages according to Pringle et al. (2010). First the amount of the gas-phase species that is able to kinetically condense onto the aerosol phase within the model time step is calculated by assuming diffusion limited condensation (Vignati et al., 2004). Then in the second stage, the partitioning between this gas phase material and the aerosol phase is performed. The partitioning calculation

is performed for all seven size modes, i.e. in each model timestep ISORROPIA is called  separately
for each of them.
According to Kakavas et al. (2022), ISORROPIA-lite features two main modifications in
its code, with regard to ISORROPIA II v2.3 (Song et al., 2018) and ISORROPIA II v1 (Fountoukis
and Nenes, 2007). First, the routines related to the stable case have been removed, since only the
metastable case is considered and all salts formed are deliquesced. However, $CaSO_4$ is the only
solid salt allowed to form, as it is considered insoluble for most atmospherically-relevant RH
values and precipitates spontaneously. Furthermore, for the calculation of binary activity
coefficients, ISORROPIA-lite uses the tabulated binary activity coefficient data for each salt from
Kusik-Meissner (Kusik and Meissner, 1978) instead of calculating them online, and includes their
temperature dependence according to Meissner and Peppas (1973). This is done by combining the
Kusik and Meissner (1978) model for specific ionic pairs with the Bromley (1973) activity
coefficient mixing rule for multicomponent mixtures. More information on this procedure, can be
found in Fountoukis and Nenes (2007). This second modification is the major contributor to the
computational speed-up provided by ISORROPIA-lite, which in an offline estimation was reported
to be around 35% (Kakavas et al., 2022). Furthermore, this feature could explain differences in
inorganic aerosol estimates with the previous version of ISORROPIA using the same aerosol state
assumption (metastable case). Another important modification is that the effect of organic aerosol
water on the inorganic semi volatile aerosol components is included. This consideration slightly
increases the aerosol pH but more significantly drives the phase partitioning towards the aerosol
phase in order to satisfy equilibrium conditions (Kakavas et al., 2022). However, this feature of
ISORROPIA-lite was not used in the present study, as the water uptake by organics is treated by
other parts of the GMXe aerosol microphysics submodel in the EMAC global model. The effects
of the secondary organic aerosol on aerosol water and nitrate partitioning are discussed by Kakavas
et al. (2023).
In the updated version of the GMXe submodel, the users have the option to select between
ISORROPIA-lite and ISORROPIA II v2.3 to perform EMAC simulations depending on the
application and the desired phase state assumption. While ISORROPIA-lite utilizes the metastable
approach exclusively, ISORROPIA II v2.3 utilizes both and has the stable approach as default.

## 3.  Evaluation of New Aerosol Thermodynamic Modules within EMAC

For reasons of clarity, from this point forward both in the main text as well as in any figure
captions, whenever different aerosol sizes are mentioned, total suspended particles (TSP) refer to
the sum of the 4 lognormal size modes of the aerosol microphysics submodel (i.e. nucleation,
Aitken, accumulation and coarse mode), fine aerosols refer to the sum of the 3 smaller size modes
(nucleation, Aitken and accumulation mode) and coarse aerosols refer to the largest size mode of
the model exclusively.

## 3.1 Comparison of ISORROPIA II v1 against ISORROPIA II v2.3 in stable mode

The first comparison aims to examine how ISORROPIA II v2.3 fares against ISORROPIA II v1 when considering solely the stable assumption, after the latter's replacement in the newer version of the EMAC model.

The differences in global daily mean surface concentrations of $NH_4^+$, $SO_4^{2-}$, mineral ions (sum of $Ca^{2+}$, $K^+$, $Mg^{+2}$), aerosol water in TSP, as well as fine and coarse aerosol $NO_3^-$ as predicted by the two versions can be seen in Figure 1. The 25[th] and 75[th] percentiles of concentration differences between the two versions for the aerosol water are below 0.2 μg m[-3] and for the remaining species they are an order of magnitude less, which translates to differences mostly below 1 % for all species. Therefore, the predictions of inorganic aerosol composition of the two versions agree exceptionally well.

In order to investigate potential differences arising in specific areas, regions affected by high nitrate concentrations were selected, i.e., Europe, the Tibetan Plateau, Eastern Asia, North America and the Middle East. The differences in daily mean coarse and fine $NO_3^-$ over these regions are shown in Figure S1. The comparison showed that the differences regarding the 25[th] and 75[th] percentiles are less than 0.05 μg m[-3] (or less than 2.5 %) between the results of the two ISORROPIA II versions for both size modes. A statistical analysis of the results reveals that all differences between the aforementioned species are on average below 3% (Table 1). Therefore, the replacement of ISORROPIA II v1 by v2.3 in the EMAC model yields only trivial differences in the predicted aerosol ionic composition and water. The following sections focus on the comparison between the results of ISORROPIA-lite against ISORROPIA II v2.3 (called ISORROPIA II hereafter for simplicity), both in stable and metastable states.

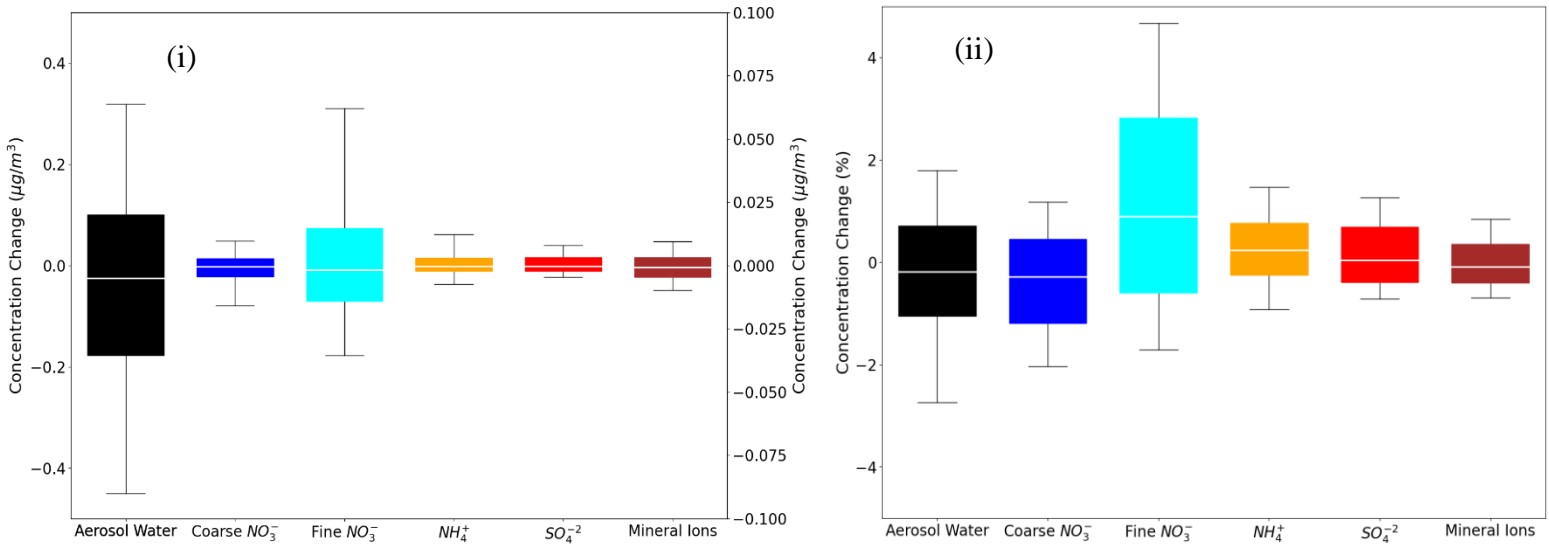


**Figure 1:** Bar chart plots depicting the 25th, 50th and 75th percentiles (box) of the i) difference and ii) fractional difference in global daily mean surface concentrations of aerosol water (left y-axis), mineral ions, $NH_4^+$ and $SO_4^{2-}$ in TSP as well as coarse and fine aerosol $NO_3^-$ (right y-axis), as predicted by EMAC using ISORROPIA II v1 and ISORROPIA II v2.3. The 10th and 90th percentiles (whiskers) for each aerosol component are also shown. Both models assume that the aerosol is at its stable state at low RH and a positive change corresponds to higher concentrations by ISORROPIA II v1.

**Table 1:** Statistical analysis of EMAC-simulated mean daily surface concentrations by employing ISORROPIA II v1 versus ISORROPIA II v2.3, both in **stable mode**. Deviations are given as ISORROPIA II v1 – ISORROPIA II v2.3.

| | Mean Difference ($\mu g/m^3$) | Normalized Mean Absolute Difference (%) |
|---|---|---|
| Coarse $NO_3^-$ | $-8 \times 10^{-4}$ | 1.8 |
| Fine $NO_3^-$ | $-0.011$ | 2.6 |
| $HNO_3$ | $-3.1 \times 10^{-4}$ | 0.7 |
| $NH_4^+$ | $-1.6 \times 10^{-4}$ | 2.0 |
| $SO_4^{2-}$ | $-0.009$ | 1.2 |
| $Na^+$ | $0.007$ | 1.6 |
| $Ca^{2+}$ | $1.7 \times 10^{-4}$ | 0.4 |
| $K^+$ | $1.1 \times 10^{-4}$ | 0.4 |
| $Mg^+$ | $1.5 \times 10^{-4}$ | 0.4 |
| $Cl^-$ | $0.040$ | 2.3 |
| $H_2O$ | $0.046$ | 1.1 |
| $H^+$ | $-2.9 \times 10^{-5}$ | 1.5 |

## 3.2 Comparison of ISORROPIA-lite against ISORROPIA II in metastable mode

The model results using ISORROPIA-lite are compared first against those using ISORROPIA II in metastable mode in order to determine whether the ISORROPIA-lite version can produce similar results with the more detailed module in EMAC, under same conditions. Figure 2 depicts the differences of the global daily mean surface concentrations of the same species that were examined before. The comparison yields differences for the 25th and 75th percentiles that are less than 0.5 $\mu g\ m^{-3}$ for the aerosol water and mostly less than 0.05 $\mu g\ m^{-3}$ for the remaining inorganic

aerosol components, which translates into differences of less than 2% for all species most of the
time.
Figure S2 shows the comparison between predicted global daily mean coarse and fine
aerosol nitrate concentrations, focusing on the regions with the higher simulated mean annual
concentrations. Across all regions, the concentration differences for both size modes are typically
lower than 0.1 μg m$^{-3}$ (or less than 3 %) and are mostly found over the Himalayan and East Asian
regions.
In Table 2, the statistics of the results for the global surface concentrations for all examined
aerosol components, reveal differences that are on average less than 7%. Therefore, ISORROPIA-
lite does provide quite similar predictions with ISORROPIA II in the EMAC model, for
simulations using the metastable state assumption.


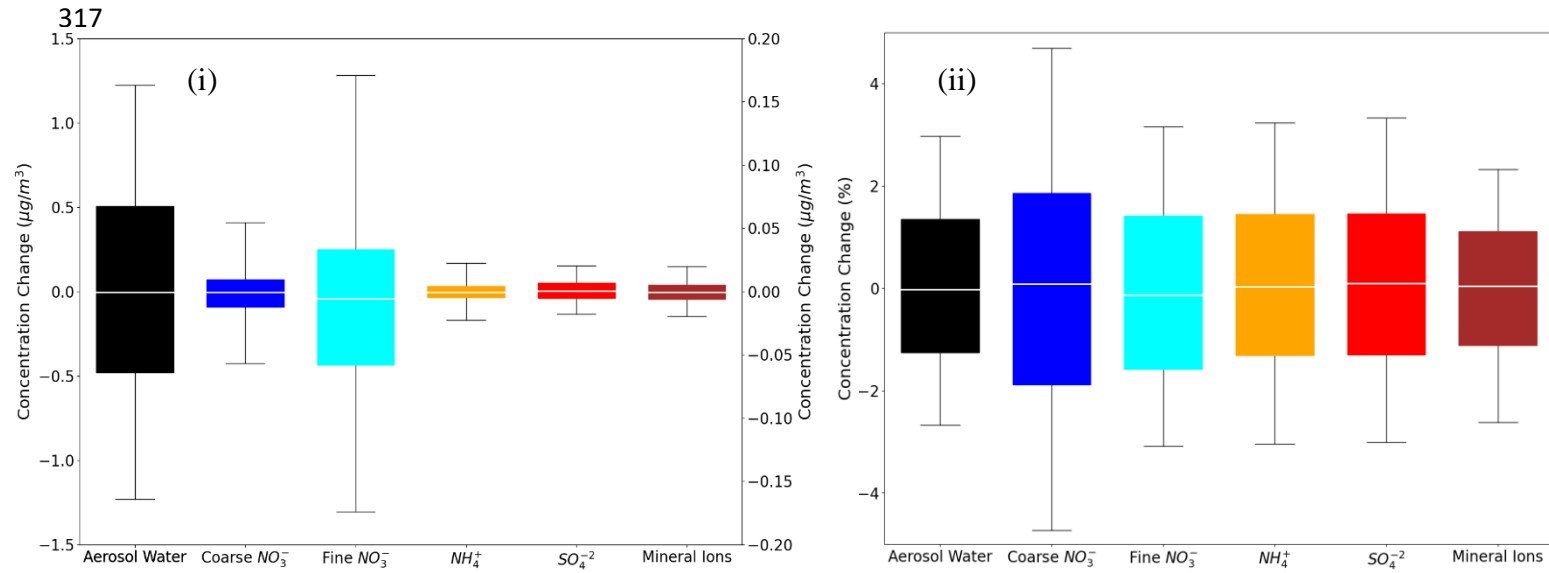


**Figure 2 :** Bar chart plots depicting the 25[th], 50[th] and 75[th] percentiles (box) of the i) difference and ii)
fractional difference in global daily mean surface concentrations of aerosol water (left y-axis), mineral ions,
NH$_4^+$ and SO$_4^{2-}$ in TSP as well as coarse and fine aerosol NO$_3^-$ (right y-axis) , as predicted by EMAC using
ISORROPIA-lite and ISORROPIA II. The 10th and 90th percentiles (whiskers) for each aerosol component
are also shown. Both models assume that the aerosol is at its metastable state at low RH and a positive
change corresponds to higher concentrations by ISORROPIA-lite.





**Table 2:** Statistical analysis of EMAC-simulated mean daily surface concentrations by employing ISORROPIA-lite versus ISORROPIA II, both in **metastable mode**. Bias is given as ISORROPIA-lite – ISORROPIA II.

| | Mean Difference ($\mu g/m^3$) | Normalized Mean Absolute Difference (%) |
|---|---|---|
| Coarse $NO_3^-$ | $-6.2 \times 10^{-4}$ | 3.5 |
| Fine $NO_3^-$ | $-3.1 \times 10^{-4}$ | 3.9 |
| $HNO_3$ | $-2.7 \times 10^{-4}$ | 2.0 |
| $NH_4^+$ | $-1.4 \times 10^{-5}$ | 3.8 |
| $SO_4^{2-}$ | $2.5 \times 10^{-3}$ | 4.0 |
| $Na^+$ | 0.011 | 6.7 |
| $Ca^{2+}$ | $2.9 \times 10^{-4}$ | 1.9 |
| $K^+$ | $1.8 \times 10^{-4}$ | 2.4 |
| $Mg^+$ | $5.8 \times 10^{-4}$ | 3.5 |
| $Cl^-$ | 0.017 | 7.0 |
| $H_2O$ | 0.035 | 1.8 |
| $H^+$ | $-8.3 \times 10^{-4}$ | 4.6 |

## 3.3 Evaluation of inorganic aerosol predictions

EMAC predictions using both ISORROPIA-lite and ISORROPIA II in stable mode for $PM_{2.5}$ ammonium, sulfate and nitrate were compared against measurements from three observational networks. The networks cover some of the most polluted areas in the Northern Hemisphere. The EPA CASTNET network (U.S. Environmental Protection Agency Clean Air Status and Trends Network) and the IMPROVE network (Interagency Monitoring of Protected Visual Environments) with 152 stations for nitrate and sulfate and 143 stations for ammonium cover the USA, with IMPROVE concerning mostly rural and/or remote areas. The EMEP network (EMEP Programme Air Pollutant Monitoring Data) includes 9 stations for nitrate and sulfate and 7 for ammonium covering the European region. Finally, the EANET network (The Acid Deposition Monitoring Network in East Asia) with 33 stations measuring all three major aerosol components covers parts of East Asia. The number of stations refers to the year 2010 which is simulated in this work.

Figure 3 depicts the differences between the model-predicted and the observed mean annual concentration values for $SO_4^{2-}$, $NH_4^+$ and $NO_3^-$ aerosols, while Tables 3, 4 and 5 contain the overall statistics for the same comparisons. Here, the mean bias (MB), mean absolute gross error (MAGE), normalized mean bias (NMB), normalized mean error (NME), and the root-mean-square error (RMSE) are calculated to assess the model performance. Starting with $SO_4^{2-}$, the model tends

to underpredict the observations but with mean bias (MB) less than -0.5 μg m$^{-3}$ for Europe and less
than -1 μg m$^{-3}$ for USA, capturing both the higher values of the Eastern US and the lower values
of the Western US. Its normalized mean error (NME) ranges from 40 to 60% being highest for the
East Asia region, which also has the highest MB of -1.65 μg m$^{-3}$(Table 3). Seasonally, the largest
biases are found during summertime over Europe and the USA and during wintertime over East
Asia (Table S4), while the same is true also for the predictions of ISORROPIA II in stable mode
exhibiting quite similar metrics (Table S1). $NH_4^+$ is much better simulated by the model over the
three regions, where the agreement with observations is high with MB values less than 0.4 μg m$^{-3}$
but with slightly higher NME values (Table 4). Over Eastern Asia, the only important disparity is
a slight underprediction of about 2 μg m$^{-3}$ around Hong Kong following the underprediction of
$SO_4^{2-}$ over the same area (Fig. 3). Seasonally, spring  is the worst period for the predictions of both
versions, while there doesn't seem to be a consistent pattern of behavior for all three regions which
perform best over different periods (Table S5 & S2). Finally, the model tends to overpredict $NO_3^-$
concentrations over the three regions with MB values less than 1 μg m$^{-3}$ albeit with high NME
values (Table 5). Over East Asia, with the exception of Hong Kong, the model overestimates the
$NO_3^-$ concentrations by about 3 μg m$^{-3}$, especially in the Wuhan and Guangzhou areas and also
around Beijing (Fig. 3). In general, besides Hong Kong, the model overpredicts the concentrations
of all three aerosol components examined here in the East Asian region. For all regions, the best
seasonal agreement between the predictions of both versions in terms of MB values is found during
the summer period, while the worst agreement occurs around the winter/spring period (Tables S6
& S3). The NME values are lowest in the summer for the USA and, surprisingly, in the winter for
Europe and East Asia,  even though this is the period with the worst MB values for these regions.
Potential explanations include the coarse grid resolution used in this work as well as issues related
to emissions (Zakoura and Pandis, 2018). It should be noted that even though the two versions
perform similarly, better performance on certain statistical metrics should not be taken as an
indication that one state assumption is more scientifically valid than the other. While a stable state
could be considered more accurate under very low humidity conditions (e.g., over remote deserts;
Karydis et al., 2016), in regions, such as those with intermediate RH and low nitrate concentration
(e.g., Northeastern US), particles are mostly in metastable state (Guo et al., 2016). However, the
two state assumptions produce very similar results in most cases, as shown here.









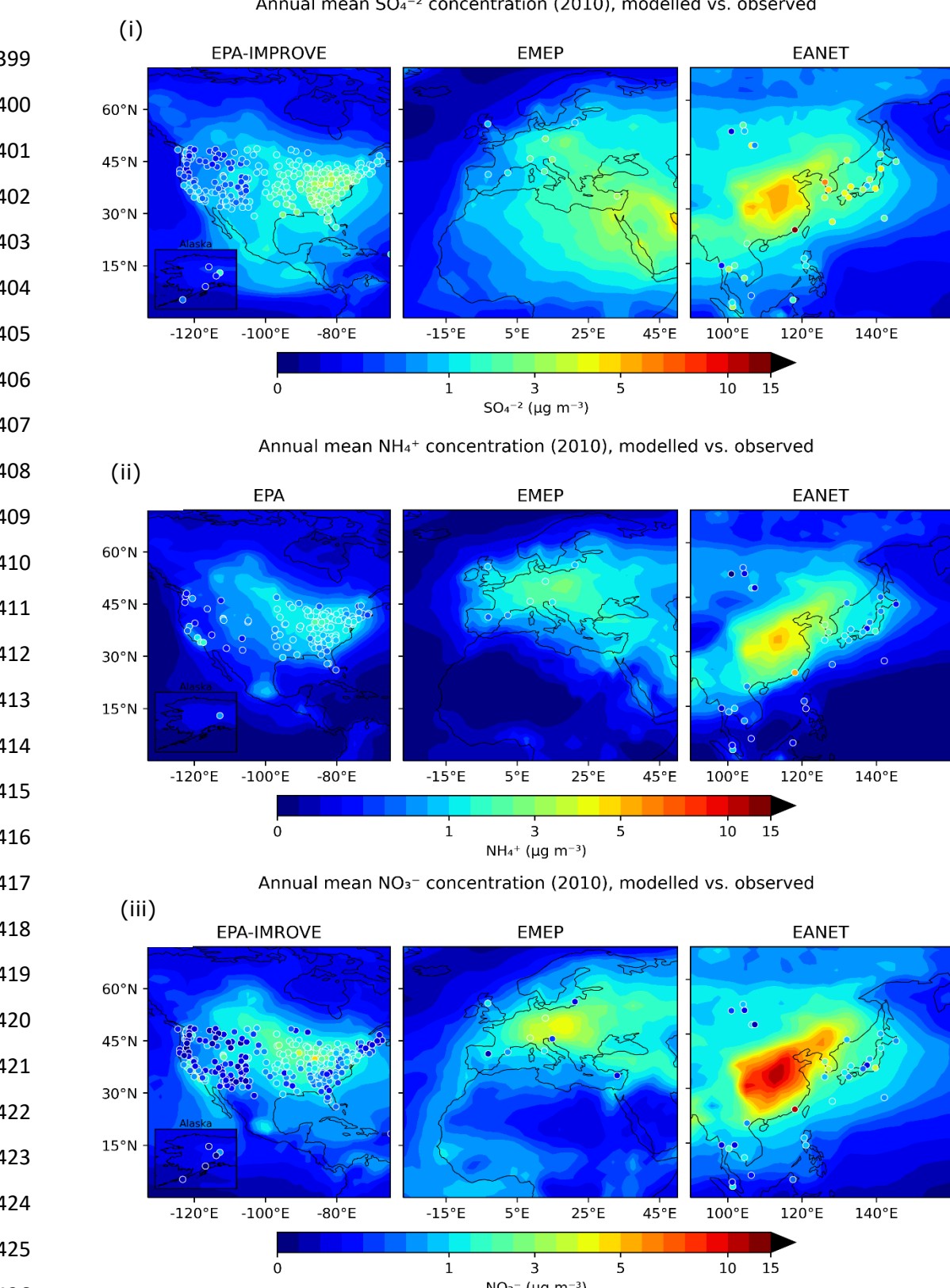

**Figure 3:** Annual mean surface concentrations of PM$_{2.5}$ i) SO$_4^{2-}$, ii) NH$_4^+$, and iii) NO$_3^-$ as simulated by EMAC using ISORROPIA-lite (shaded contours) versus observations of the same species from the IMPROVE, EMEP and EANET networks (colored circles).


**Table 3:** Statistical evaluation of EMAC predicted surface concentrations of $PM_{2.5}$ $SO_4^{2-}$ using ISORROPIA-lite against observations during 2010.


| Network | Number of datasets | Mean Observed ($\mu g\ m^{-3}$) | Mean Predicted ($\mu g\ m^{-3}$) | MAGE ($\mu g\ m^{-3}$) | MB ($\mu g\ m^{-3}$) | NME (%) | NMB (%) | RMSE ($\mu g\ m^{-3}$) |
|---|---|---|---|---|---|---|---|---|
| EPA | 1791 | 2.18 | 1.28 | 0.92 | -0.90 | 42 | -38 | 0.93 |
| IMPROVE | 1526 | 1.02 | 0.92 | 0.47 | -0.10 | 46 | -11 | 0.73 |
| EMEP | 108 | 1.71 | 1.27 | 0.75 | -0.44 | 44 | -26 | 0.91 |
| EANET | 353 | 3.19 | 1.54 | 1.95 | -1.65 | 61 | -51 | 2.46 |


**Table 4:** Statistical evaluation of EMAC predicted surface concentrations of $PM_{2.5}$ $NH_4^+$ using ISORROPIA-lite against observations during 2010.

| Network | Number of datasets | Mean Observed ($\mu g\ m^{-3}$) | Mean Predicted ($\mu g\ m^{-3}$) | MAGE ($\mu g\ m^{-3}$) | MB ($\mu g\ m^{-3}$) | NME (%) | NMB (%) | RMSE ($\mu g\ m^{-3}$) |
|---|---|---|---|---|---|---|---|---|
| EPA | 1660 | 1.01 | 1.01 | 0.50 | 0.00 | 49 | 0 | 0.72 |
| IMPROVE | - | - | - | - | - | - | - | - |
| EMEP | 84 | 1.08 | 1.44 | 0.63 | 0.36 | 59 | 34 | 0.75 |
| EANET | 360 | 0.93 | 1.25 | 0.69 | 0.32 | 74 | 34 | 1.25 |


**Table 5:** Statistical evaluation of EMAC predicted surface concentrations of $PM_{2.5}$ $NO_3^-$ using ISORROPIA-lite against observations during 2010.

| Network | Number of datasets | Mean Observed ($\mu g\ m^{-3}$) | Mean Predicted ($\mu g\ m^{-3}$) | MAGE ($\mu g\ m^{-3}$) | MB ($\mu g\ m^{-3}$) | NME (%) | NMB (%) | RMSE ($\mu g\ m^{-3}$) |
|---|---|---|---|---|---|---|---|---|
| EPA | 1762 | 1.39 | 1.87 | 1.06 | 0.48 | 76 | 42 | 1.65 |
| IMPROVE | 1526 | 0.42 | 1.18 | 0.82 | 0.76 | 194 | 175 | 1.15 |
| EMEP | 108 | 1.15 | 1.91 | 1.25 | 0.76 | 109 | 66 | 1.66 |
| EANET | 372 | 1.32 | 2.27 | 1.33 | 0.95 | 101 | 72 | 2.17 |





## 3.4 Computational speed-up metrics

The computational efficiency and speed-up that ISORROPIA-lite provides compared to ISORROPIA II in both stable and metastable modes were quantified. Table 6 contains the total number of time steps that the EMAC model performed for the same simulation period (i.e., 24 h of CPU-time using 16 nodes) as well as the real time that was needed per individual time step, for each ISORROPIA version. The metrics shown in Table 6 concern the average value of each quantity, along with the corresponding standard deviation, resulting from a total of 18 simulations (6 for each version). From the difference in the real time required by the model to execute each individual time step, the speed-up of ISORROPIA-lite was found to be just above 3% compared to ISORROPIA II in metastable mode and almost 5 % compared to ISORROPIA II in stable mode. These values are, as expected, lower than the improvement in the computational efficiency that the ISORROPIA-lite version provides compared to the original version, as found in the offline evaluation, because EMAC contains several other modules that are quite computationally expensive. For example, the gas-phase chemistry (MECCA submodel) as well as wet deposition and liquid phase chemistry (SCAV submodel) are responsible for two thirds of the total computational cost of the global model. As a comparison, the offline speed-up that ISORROPIA-lite provided was calculated to be 35% and when utilized in the regional model PMCAMx 3D it was found to be 10% (Kakavas et al., 2022).

**Table 6:** Total number of time steps that EMAC executed in 24 hours of running time and number of seconds needed for each time step, utilizing ISORROPIA-lite and ISORROPIA II (both in Stable & Metastable). The computational speed-up refers to how much quicker (in %) the process is executed by ISORROPIA-lite in comparison to the previous version in both modes.

| Simulation | # Time Steps | # Seconds per Timestep | Computational Speed-Up (%) |
|---|---|---|---|
| ISORROPIA-lite | $78{,}193 \pm 116$ | $1.10 \pm 0.002$ | - |
| ISORROPIA II v2.3 (Metastable) | $75{,}720 \pm 242$ | $1.14 \pm 0.003$ | $3.3 \pm 0.3$ |
| ISORROPIA II v2.3 (Stable) | $74{,}599 \pm 169$ | $1.16 \pm 0.003$ | $4.8 \pm 0.3$ |

## 4. Comparison of ISORROPIA-lite Against ISORROPIA II in Stable Mode

In this section, we present a comparison of the ISORROPIA-lite results in metastable mode against those of the ISORROPIA II results in stable mode. Both versions are now available in the latest version of the EMAC model, and the user has the option to utilize either one. While ISORROPIA-lite always assumes metastable aerosols, ISORROPIA II assumes stable aerosols by default. This comparison is done in an attempt to quantify the effects of using the metastable case in global atmospheric simulations, and to identify the regions and conditions under which the two assumptions have any significant differences. Some discrepancies are expected due to the different physical state of aerosols at low RH, however, the choice between a stable state and a metastable state should not be considered obvious. For example, Fountoukis et al. (2009) and Karydis et al. (2010) have shown that the stable assumption is in better agreement with observations under conditions where RH is consistently below 50%. On the other hand, Ansari and Pandis (2000) emphasize that the metastable assumption must be considered for regions characterized by intermediate RH and low pollutant concentrations (in this case of $NO_3^-$), while there are no significant differences between the two assumptions over regions with high concentrations. Here, differences in the calculated aerosol acidity by the two modules are also investigated.

### 4.1 Spatial variability of mean annual aerosol concentrations

For sulfate in TSP the predicted maximum annual average concentration was 7 µg m$^{-3}$ found over East Asia highlighting the large anthropogenic impact over that region, while it was also high (> 5 µg/m$^3$) in India, Europe, and the Middle East in both simulations (Fig. 4i). Absolute differences for sulfate in TSP were lower than 0.15 µg m$^{-3}$ (< 3%) and found mainly over the polluted northern hemisphere (mainly East USA & Europe) with slightly higher values simulated by ISORROPIA II (Fig. 4ii). This is most likely related to the also higher $NO_3^-$ aerosol predictions by ISORROPIA-lite over the same regions (see below & Fig. 4viii). The higher $SO_4^{2-}$ aerosol concentrations estimated by ISORROPIA II over the Middle East region are mainly due to changes in wet deposition induced by the different physical state of the aerosol due to the higher water content by ISORROPIA-lite. The simulated concentrations of $NH_4^+$ in TSP had maximum annual average values of 6 µg m$^{-3}$ and were found mainly over East Asia, especially around the greater Beijing and Wuhan areas, while India and Europe also exhibited high mean annual values for TSP $NH_4^+$ (> 3 µg m$^{-3}$) (Fig. 4iii). The absolute differences for $NH_4^+$ in TSP between the two model versions are higher over the Himalayan and East Asian regions (in favor of ISORROPIA II) but apparently weaker over USA, the Middle East and Africa (ISORROPIA-lite predicts higher values), although never higher than 0.5 µg m$^{-3}$ (< 5%) (Fig. 4iv). Regarding aerosol $NO_3^-$ concentrations in the coarse mode the maximum annual average of 6 µg m$^{-3}$ was predicted at the Arabian Peninsula (Fig. 4v), while in the fine mode the maximum annual average value of 11 µg m$^{-3}$ was predicted over the metropolitan areas of Wuhan and Guangzhou (Fig. 4vii). Other high annual average concentrations of fine aerosol $NO_3^-$ are found in the Tibetan Plateau and most prominently in heavy industrial regions such as East US, Eastern Asia and Europe (exceeding 4 µg m$^{-3}$ in most of these areas) with the latter two regions contributing high annual average concentrations in the coarse mode as well.

The absolute differences for coarse $NO_3^-$ were similar in magnitude to those of $NH_4^+$ in TSP with the Middle East yielding higher values by ISORROPIA-lite while the opposite is true for Europe and East USA (Fig. 4vi). The absolute differences for fine $NO_3^-$ are higher than those for coarse $NO_3^-$ reaching up to 1.75 µg m$^{-3}$ mainly over the Tibetan Plateau (~ 30%) with ISORROPIA II predicting the higher values (Fig. 4vii). Higher nitrate concentrations were also predicted by ISORROPIA II mainly close to the West coast of South America and North of Atacama Desert. Around those regions as well as the Tibetan Plateau, the relative humidity is often below 50% and 30% respectively (see Fig. 8) and the metastable assumption results in lower nitrate concentrations, in agreement with the findings of Ansari and Pandis (2000). At the same time, ISORROPIA II predicts a higher aerosol fraction for $NO_3^-$ (up to 10%) for the West coast of South America and the Tibetan Plateau. This is not the case for East Asia (Fig. 5ii) although the low sulfate/nitrate ratio of that region, results to an excess of available $NH_3$ to react with $HNO_3$ and form ammonium nitrate that would justify the higher fine mode nitrate concentrations by the stable case of ISORROPIA II (Ansari and Pandis, 2000). A higher $NO_3^-$ aerosol fraction (up to 10%) in the Middle East was exhibited by ISORROPIA-lite (Fig. 5ii). This area is characterized by increased mineral ion concentrations and high sulfate to nitrate ratios (Karydis et al., 2016) which led to higher coarse mode nitrate predictions by the metastable case (Ansari and Pandis, 2000), although the maximum difference was only 0.6 µg m$^{-3}$ (Fig. 4vi, 4viii). The differences in coarse and fine $NO_3^-$ among the two versions did not display any strong seasonality as they were only slightly higher during autumn (for East Asia) and winter (for India-Himalayas) (not shown). A comparison of the simulated aerosol concentrations at higher altitudes can be found in Figure S3, where the zonal mean annual average concentrations as well as their absolute differences between the two model versions are depicted. The deviations between the results of the two ISORROPIA versions are becoming smaller as the air masses move higher in the atmosphere, until they are practically identical at altitudes above 700hPa. Regarding the behavior of the mineral ions of $Ca^{2+}$, $K^+$, and $Mg^{2+}$ the majority of high concentrations are found around the largest desert regions of the Sahara, Gobi, Atacama and Namib deserts (Figure S4), with $Ca^{2+}$ being evidently the most dominant across all minerals. Furthermore, the absolute difference maps (Fig. S4) show minimal differences in mean annual surface concentrations (mostly less than 0.5 µg m$^{-3}$) between the simulations from the two model versions. This is also reflected in the comparison of zonal mean annual average concentrations and their differences, as shown in Figure S5.

In the heavily polluted regions (particularly East USA, Europe and East Asia), the particulate $NO_3^-$ dominates compared to the gas phase $HNO_3$ (Fig. 5i). The fine-mode fraction of the particulate nitrate burden is bigger than the coarse-mode fraction over Eastern Asia, India, Europe, and Eastern USA, while in the large desert areas of the Middle East and the Sahara most of the particulate $NO_3^-$ exists in the coarse mode (Fig. 5iii). The aerosol water fraction is low (<30%) across the most arid regions of Sahara, Atacama, Namib and Gobi, while Europe has the highest continental average aerosol water content in the Northern Hemisphere polluted regions (Fig. 5v). ISORROPIA-lite predicts higher average aerosol water concentration globally since the particles cannot form solids, and the salts remain in a supersaturated metastable solution (Fig. 5vi).

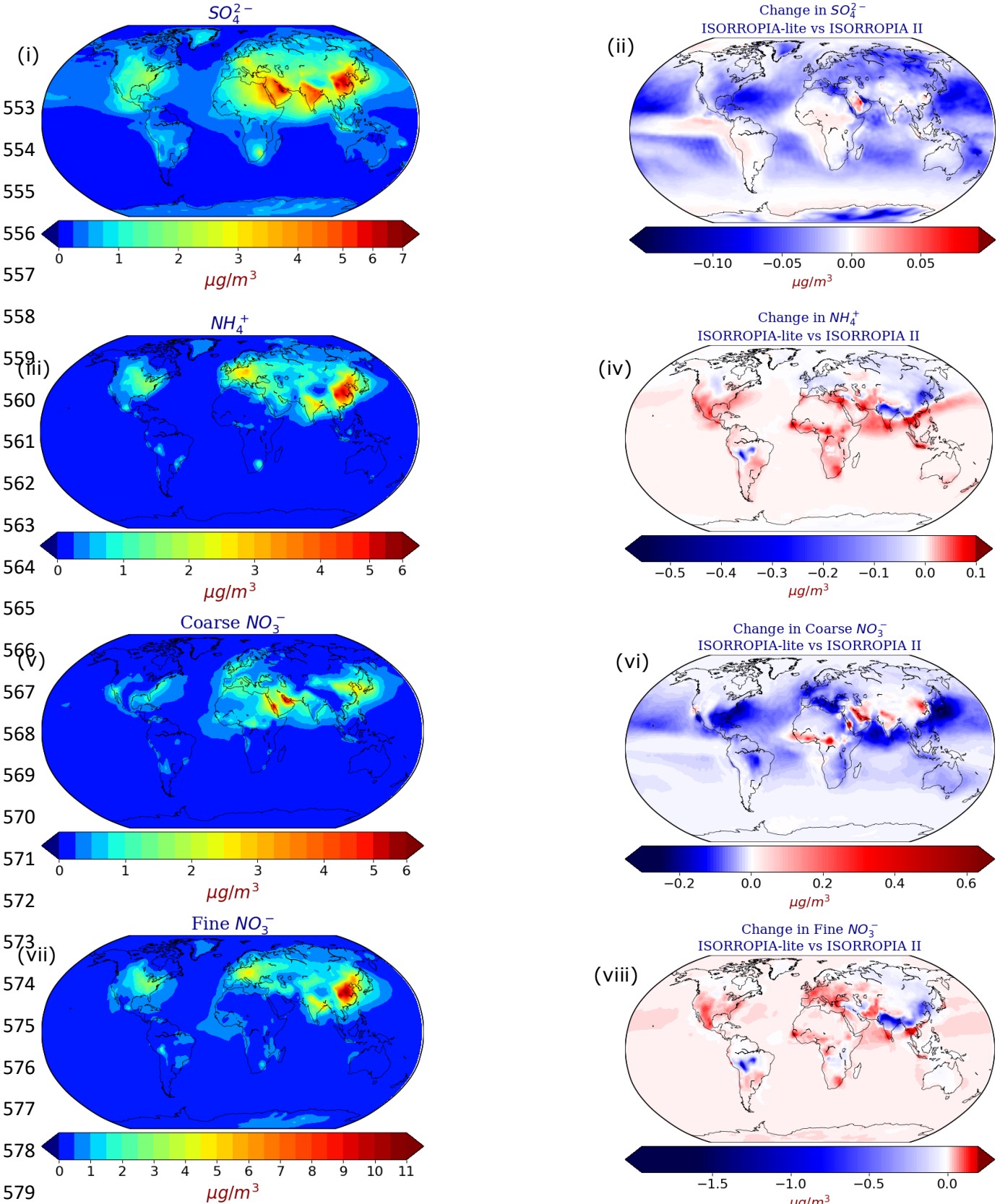

**Figure 4 :** Annual mean surface concentrations of i) $SO_4^{2-}$ and ii) $NH_4^+$ in TSP, iii) coarse and iv) fine aerosol $NO_3^-$ as predicted by EMAC using ISORROPIA-lite. Change of the annual mean EMAC-simulated surface concentration of v) $NH_4^+$ and vi) $SO_4^{2-}$ in TSP, vii) coarse and viii) fine aerosol $NO_3^-$ after employing ISORROPIA II. Positive values in red indicate higher concentrations by ISORROPIA-lite. The models assume different aerosol states.

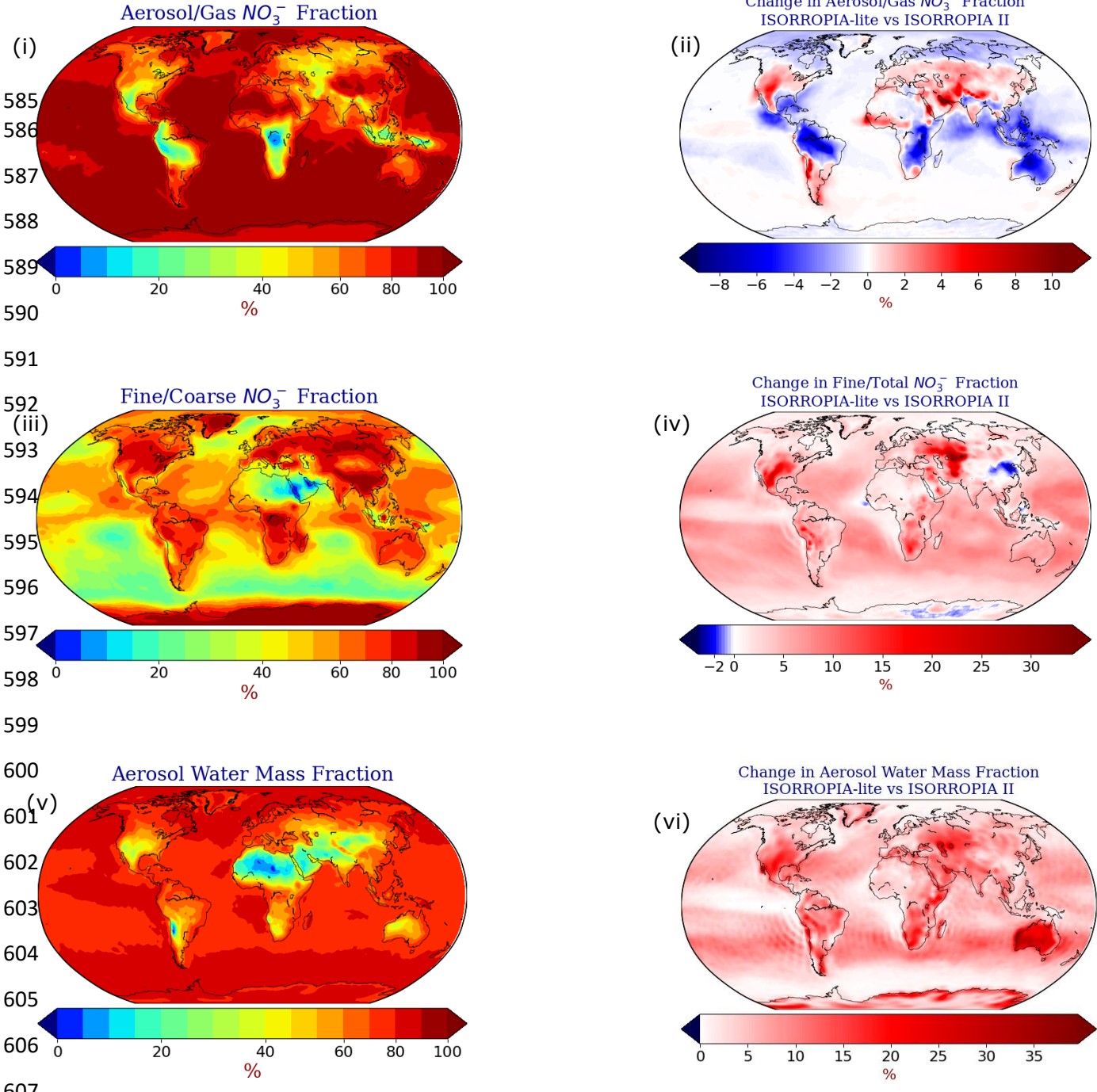

**Figure 5 :** Annual mean surface fractions of i) aerosol/gas NO$_3^-$, ii) fine/total-aerosol NO$_3^-$ and iii) aerosol water mass as calculated by EMAC using ISORROPIA-lite. Change of the annual mean EMAC-simulated surface fractions of aerosol/gas iv) NO$_3^-$, v) fine/total-aerosol NO$_3^-$, and vi) aerosol water mass after employing ISORROPIA II. Positive values in red indicate higher fractions by ISORROPIA-lite. The models assume different aerosol states.

The absolute differences in global daily mean concentrations are mostly less than 0.3 μg
$m^{-3}$ for all species ($NH_4^+$, $SO_4^{2-}$ and Mineral Cations in TSP as well as coarse and fine aerosol $NO_3^-$
) except aerosol water in TSP (Figure 6). In that case the absolute differences for the 25th and 75th
percentiles are less than 5 μg $m^{-3}$. This translates to fractional differences for the 25th and 75th
percentiles mostly below 20 %  for aerosol water in TSP and coarse $NO_3^-$ aerosol, and mostly
below 5% for all the remaining species.

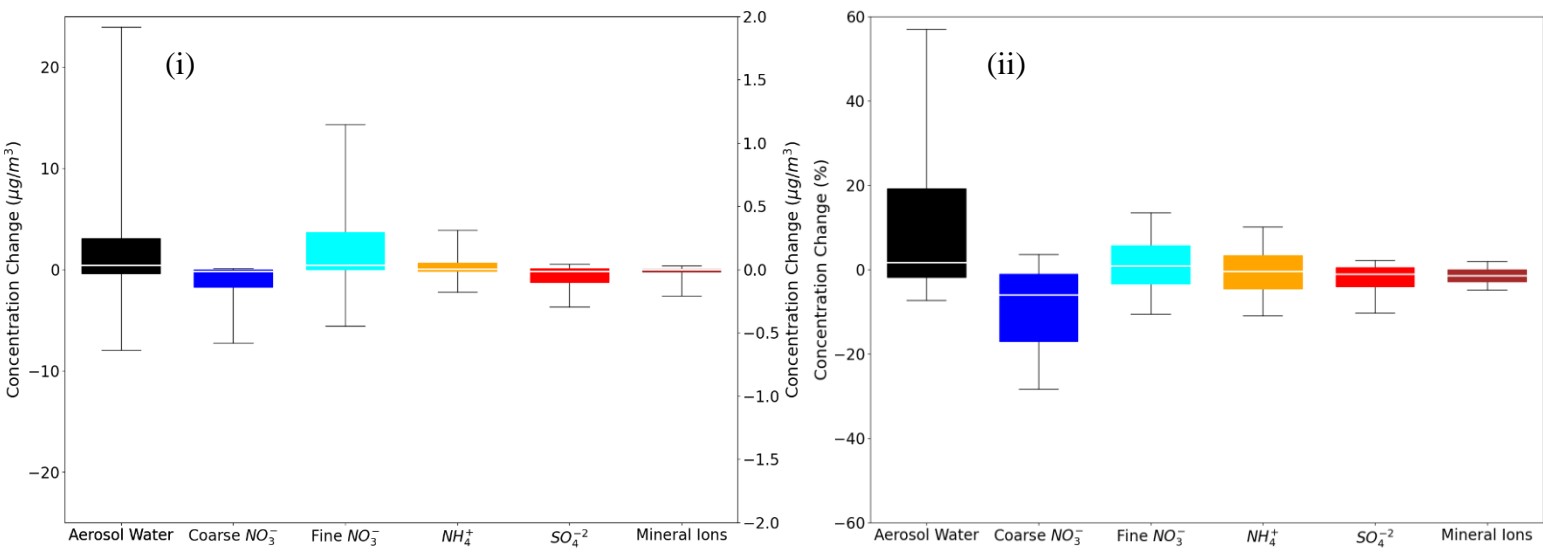


**Figure 6 :** Bar chart plots depicting the 25th, 50th and 75th percentiles (box) of the i) difference and ii)
fractional difference in global daily mean surface concentrations of aerosol water (left y-axis), mineral ions,
$NH_4^+$ and $SO_4^{2-}$ in TSP as well as coarse and fine aerosol $NO_3^-$ (right y-axis) , as predicted by EMAC using
ISORROPIA-lite and ISORROPIA II. The models assume different aerosol states at low RH and a positive
change corresponds to higher concentrations by ISORROPIA-lite.











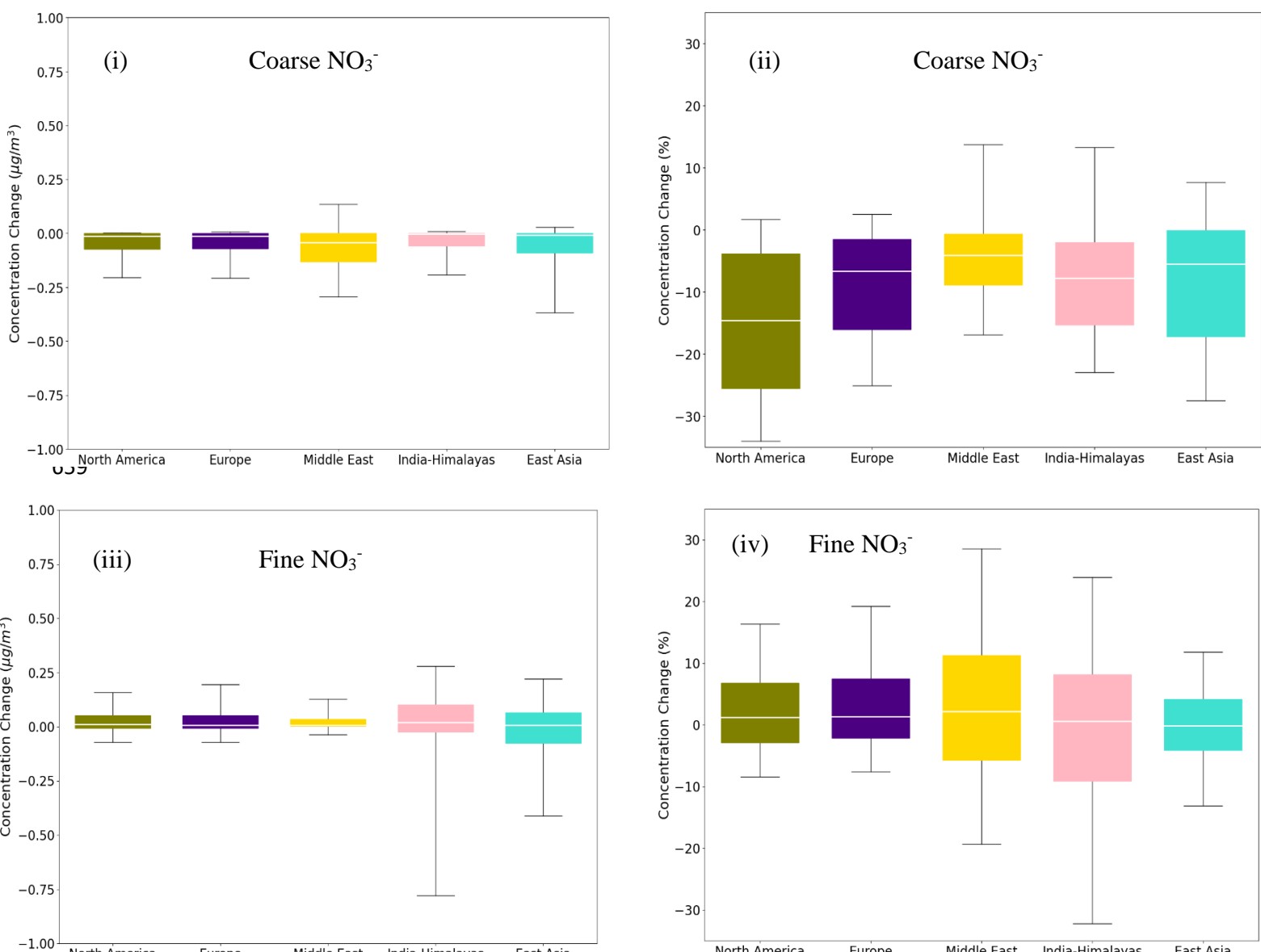


**Figure 7:** Bar chart plots depicting the 25[th], 50[th] and 75[th] percentiles (box) of the difference in the global
daily mean surface concentrations of i) coarse and ii) fine aerosol $NO_3^-$ for the regions of North America,
Europe, Middle East, India-Himalayas and East Asia, as predicted by EMAC using ISORROPIA-lite and
ISORROPIA II. The fractional differences in global daily mean surface concentrations of iii) coarse and
iv) fine aerosol $NO_3^-$ for the same regions are also shown. The models assume different aerosol states at
low RH and a positive change corresponds to higher concentrations by ISORROPIA-lite.

The model results in the regions with the highest mean annual loads of fine and coarse aerosol
$NO_3^-$ concentrations (see Section 3.1) as well as the most significant differences in estimated
aerosol water and aerosol acidity (see Section 4.3), were further analyzed to determine whether
the phase state assumption has a large effect on simulated aerosol nitrate formation (Figure 7). For
both coarse and fine daily mean $NO_3^-$ concentrations, Europe and North America are clearly the
regions with the smallest differences between the two versions. On the other hand, East Asia and
especially the India-Himalayas regions are areas where the differences are the highest with
ISORROPIA II predicting higher fine aerosol $NO_3^-$ concentrations while in the Middle East,
ISORROPIA-lite is predicting higher coarse mode aerosol $NO_3^-$ concentrations. However, even
for these areas the differences are typically below 0.25 μg m$^{-3}$ (25[th] and 75[th] percentiles) with the

higher differences not exceeding 0.8 μg m$^{-3}$ ($10^{th}$ and $90^{th}$ percentiles). This translates to fractional differences below 25 % ($25^{th}$ and $75^{th}$ percentiles) for all regions, reaching up to 30 % ($10^{th}$ and $90^{th}$ percentiles) mainly in the Tibetan Plateau and the Middle East.

Table 7 contains the statistics for the comparisons of the global daily average surface concentrations calculated by the two simulations. While all the aerosol component concentrations, except for aerosol water, are higher for ISORROPIA II, the differences are still quite low. Furthermore, despite the different aerosol phase state assumption by the two versions, the normalized mean absolute difference remains low for all species (on average < 11 %) except HNO$_3$. The overall statistics support the conclusion that on the global scale, the phase state assumption for low RH does not have a significant impact on the predicted tropospheric aerosol load. More specifically, ISORROPIA-lite produces a slightly higher tropospheric burden for aerosol NO$_3^-$ than ISORROPIA II (0.875 Tg versus 0.861 Tg, respectively) while the opposite was the case for HNO$_3$ (0.921 Tg versus 0.935 Tg). The higher burden of ISORROPIA-lite is due to the fact that the higher aerosol water content favors the partitioning of HNO$_3$ to the particulate phase.

**Table 7:** Statistical analysis of EMAC-simulated mean daily surface concentrations by employing ISORROPIA-lite in **metastable mode** versus ISORROPIA II in **stable mode**. Bias is given as ISORROPIA-lite – ISORROPIA II.

| | Mean Difference ($\mu g/m^3$) | Normalized Mean Absolute Difference (%) |
|---|---|---|
| Coarse $NO_3^-$ | -0.026 | 9.1 |
| Fine $NO_3^-$ | -0.044 | 9.8 |
| $HNO_3$ | -0.002 | 10.3 |
| $NH_4^+$ | $-1.8 \times 10^{-4}$ | 8.0 |
| $SO_4^{2-}$ | -0.020 | 4.8 |
| $Na^+$ | -0.081 | 8.6 |
| $Ca^{2+}$ | -0.005 | 1.7 |
| $K^+$ | -0.002 | 1.8 |
| $Mg^+$ | -0.002 | 1.7 |
| $Cl^-$ | -0.120 | 9.4 |
| $H_2O$ | 2.717 | 10.8 |
| $H^+$ | $-4.7 \times 10^{-4}$ | 6.1 |
| pH Accumulation | -0.06 (pH) | 2.3 |
| pH Coarse | 0.03 (pH) | 2.3 |

## 4.2 Relative humidity dependent behavior of $NO_3^-$ aerosols

The dependence of the differences in nitrate predictions on relative humidity was examined both for fine and coarse particles (Figure 8). The differences between ISORROPIA II and ISORROPIA-lite are higher at intermediate RH ranging from 20% to 60% being more evident in the fine mode aerosol $NO_3^-$ and for high annual mean concentrations of coarse mode aerosol $NO_3^-$ (> 4 μg m$^{-3}$). In this RH range, solid salts can precipitate when the stable equilibrium state is assumed (Seinfeld and Pandis, 2016), while in the metastable state all these salts remain dissolved in water. A region that has often RH in the 20 - 60% range is the Tibetan Plateau which leads to discrepancies of the fine mode particulate nitrate predictions of the two models in this area, while higher coarse mode particulate nitrate concentrations are predicted by ISORROPIA-lite in the Middle East, an area that is also often characterized by intermediate RH. The differences found for coarse mode particulate nitrate in the higher RH ranges of 60 – 100 %, can account for the respective differences that occurred in areas characterized by such RH values (East USA, Europe and East Asia) but concern lower annual mean concentration values (< 3 μg m$^{-3}$).

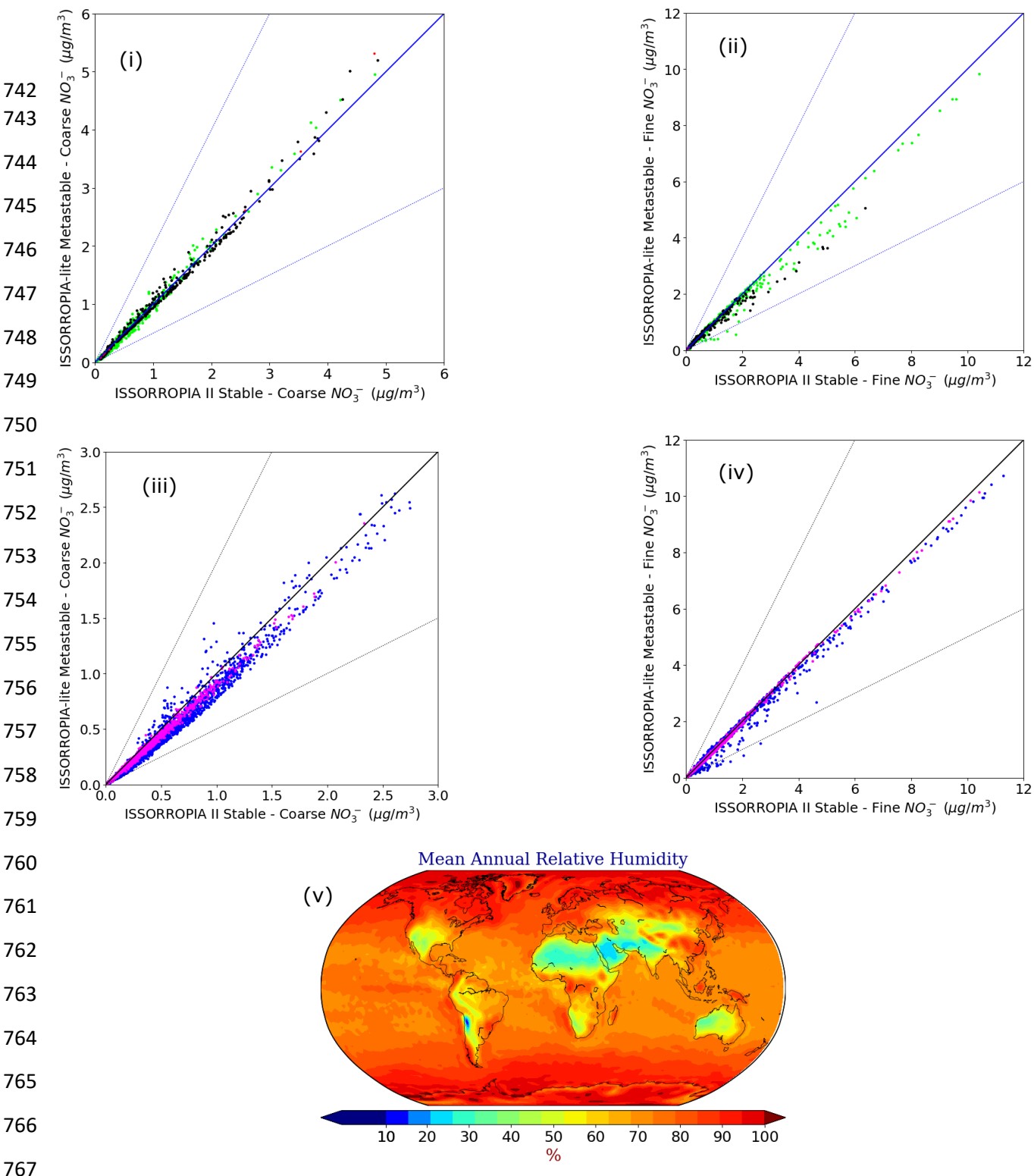

**Figure 8:** Scatterplots comparing the annual mean surface concentrations of coarse (i, iii) and fine aerosol $NO_3^-$ (ii, iv) for relative humidity ranges of 20-60 % (i, ii) and 60-100 % (iii, iv) as predicted by EMAC using ISORROPIA-lite versus ISORROPIA II. The models assume different aerosol states at low RH. Black points represent the 20-40 % RH range, green points the 40-60 % range, blue points the 60-80 % range and pink points the 80-100 % range. v) Mean annual relative humidity as calculated by EMAC using ISORROPIA-lite.

## 4.3 Comparison of the estimated aerosol acidity

The estimated aerosol acidity by the two model versions was compared separately for the accumulation and coarse size modes. This comparison aims at verifying the credibility of the estimated inorganic aerosol acidity of ISORROPIA-lite, as the first results of its implementation in the EMAC model are presented here. Since this capability is well established for ISORROPIA II (Karydis et al., 2021), it is of interest to examine any potential, but otherwise expected, differences between the two versions. The pH was computed for the fine and coarse particles, by:

$$pH = -log_{10}(\frac{[H^+]}{[H2O]}) \tag{1}$$

The calculations were performed neglecting the water associated with the organic fraction of aerosols, as they are handled by other parts of the aerosol microphysics submodel GMXe. The average pH was calculated based on the instantaneous $H^+$ and $H_2O$ values estimated every 5 hours. This is because utilizing daily average values for $H^+$ and $H_2O$ can result in a low-biased predicted pH of ~2 units globally (Karydis et al., 2021). The 5 hour interval provides a frequent output of values at different times of the day to account for the diurnal variability of pH, since a selection of 6 or 8 hour intervals would result in instantaneous $H_+$ and $H_2O$ values at identical times on different days. pH calculations are performed only in cases where there is enough water in the aerosol (instantaneous values exceeding 0.05 μg m$^{-3}$).

ISORROPIA-lite predicts slightly more acidic particles mainly in the coarse mode (Fig. 9iv). The most significant differences (up to 1 unit) in that size range are located over the Middle East and Arabian Peninsula, while smaller differences can be found in limited parts of the Himalayan and the East Asian regions as well as West USA and the Amazon Basin. These regions are characterized by high mineral cation concentrations and/or low RH. Therefore, the stable state results in increased pH values due to the precipitation of insoluble salts out of the aqueous phase. On the other hand, in the metastable state all anions remain in the aqueous phase lowering the particle pH. Differences in accumulation mode particle acidity are not as high (Fig. 9ii). ISORROPIA-lite predicts that accumulation mode particles over heavy industrialized regions such as Southeast Asia, Europe and Eastern USA are moderately acidic with mean pH values in the range of 4 - 5 while exhibiting alkaline behavior in desert areas where the increased levels of mineral ions elevate the pH above 7 (Fig. 9i). Coarse mode particles are in general more alkaline than those in the accumulation mode, with a few exceptions over east US, central Europe, north India and SE Asia (Fig. 9iii). These regions are characterized by high $NH_3$ concentrations from agricultural activities.

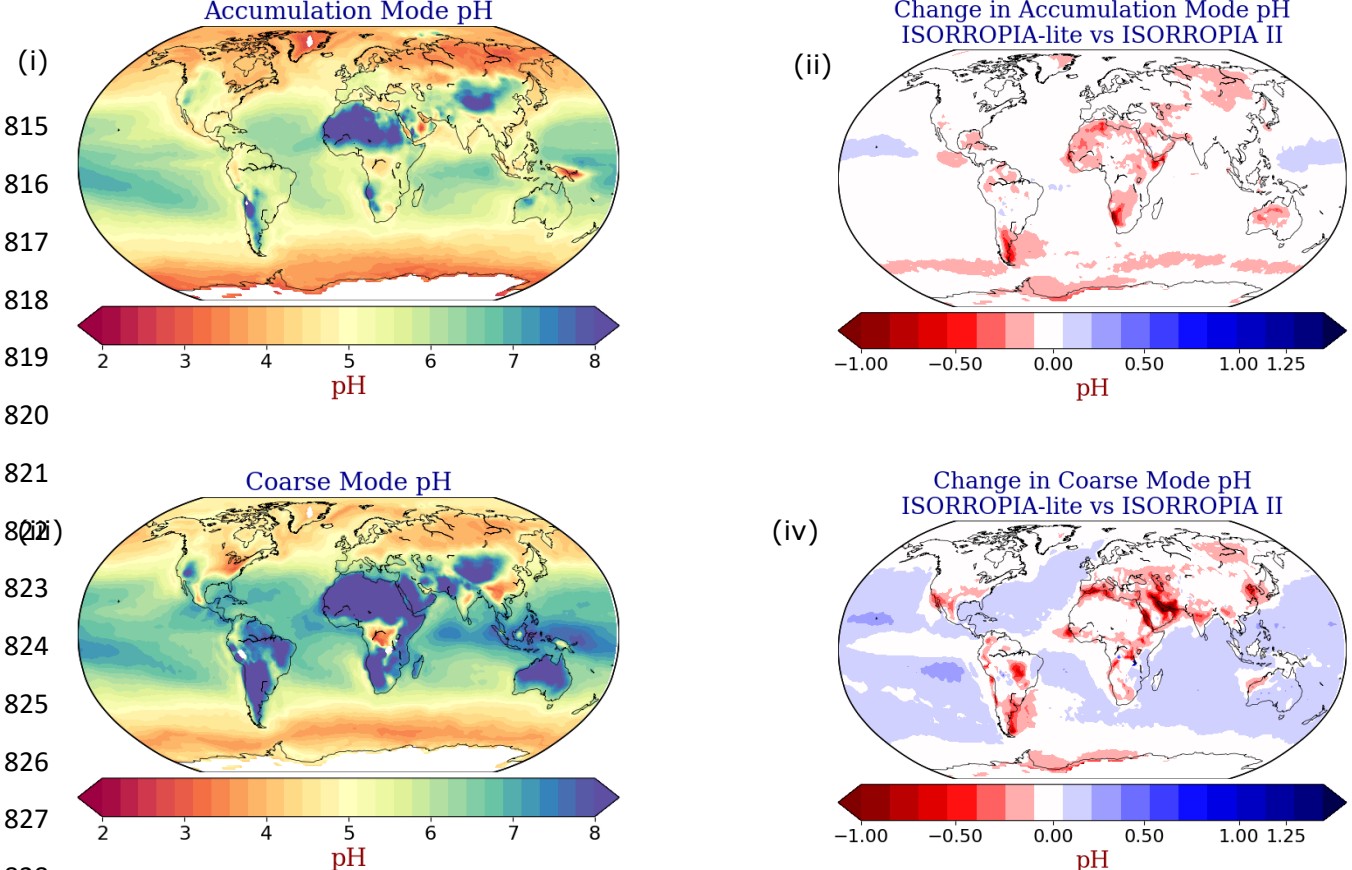


**Figure 9:** Annual mean EMAC-simulated i) accumulation and ii) coarse mode aerosol pH using
ISORROPIA-lite. Change of the annual mean EMAC-simulated iii) accumulation and iv) coarse mode
aerosol pH after using ISORROPIA II, with negative values in red indicating lower pH by ISORROPIA-
lite. The models assume different aerosol states.

A sensitivity test was performed by reducing all NH$_3$ emissions by half to investigate if
there is a buffering mechanism that controls the pH of the accumulation mode particles more than
in the coarse mode. Figure 10 shows the difference of the mean annual calculated aerosol pH
between the base case (NH$_3$ emissions present) and the sensitivity case (half NH$_3$ emissions).
When NH$_3$ emissions are switched off, the pH of fine PM decreases by up to 3 units and the
particles become a lot more acidic (Fig. 10i). For the coarse mode this effect is not that strong (pH
reduction of up to 1.5 units) (Fig. 10ii). As expected, this buffering mechanism is mainly observed
across the aforementioned regions where NH$_3$ concentrations are high, but also over areas affected
by natural NH$_3$ emissions. This is consistent with the results of Karydis et al. (2021) who found
that in the absence of NH$_3$, aerosol particles would be extremely acidic in most of the world.
The differences in the accumulation mode pH calculated by ISORROPIA-lite and
ISORROPIA II are extremely small (i.e., mean difference of 0.06 pH units or 2.3%), and even
smaller for coarse mode pH (Table 7), indicating an overall good agreement between the two
model versions.


Change in Accumulation Mode pH in the presence of *NH₃*

Change in Coarse Mode pH in the presence of *NH₃*

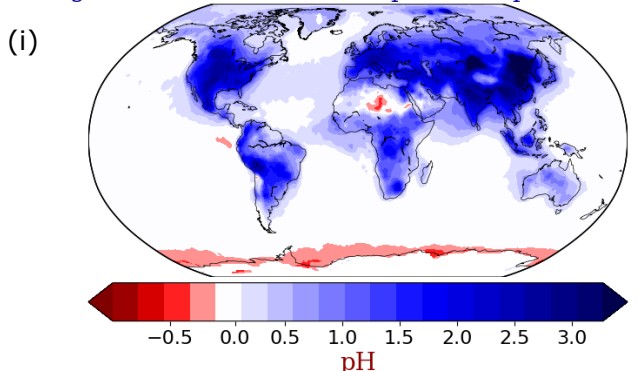
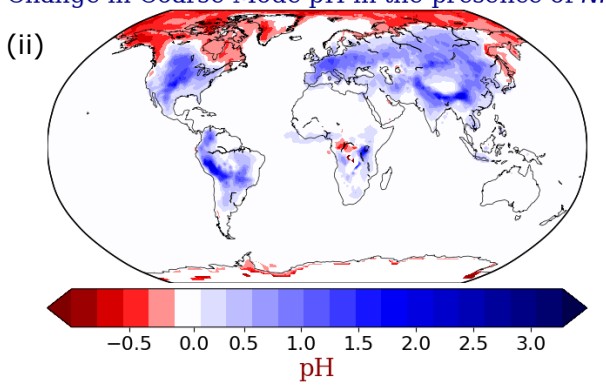

**Figure 10:** Absolute change of the annual mean EMAC-simulated i) accumulation and ii) coarse mode aerosol pH using ISORROPIA-lite after reducing the $NH_3$ emissions by half. Positive values in blue indicate higher aerosol pH when $NH_3$ is present.

## 5. Conclusions

This study presents the first results of the implementation of the ISORROPIA-lite thermodynamic module in the EMAC global chemistry and climate model, and is compared to the previous version, ISORROPIA II v2.3, after the latter has successfully replaced ISORROPIA II v1 to improve pH predictions close to neutral conditions.

The results of ISORROPIA II versions 1 and 2.3 both in stable mode, had insignificant differences (<3%) concerning the global predictions of $NH_4^+$, $SO_4^{2-}$, mineral ions and aerosol water in TSP concentrations as well as fine and coarse mode aerosol $NO_3^-$. The comparison of results from ISORROPIA-lite against ISORROPIA II v2.3 in metastable mode, showed also negligible differences (<7%) between all the examined aerosol components on a global scale. The comparison of the ISORROPIA-lite results for $PM_{2.5}$ $NH_4^+$, $SO_4^2$, and $NO_3^-$ versus observations from the IMPROVE, EMEP and EANET networks reveals that East Asia is the area with the largest discrepancies. There was satisfactory agreement in Europe and over the US for $NH_4^+$ and $SO_4^{2-}$, while ISORROPIA-lite predicted lower concentrations around Hong Kong with a maximum difference of 1.5 µg m$^{-3}$ (~20 %) for these two species. For $NO_3^-$, the discrepancy was up to 3 µg m$^{-3}$ (~30 %) in the same region, while a difference of about 1.5 µg m$^{-3}$ (~25 %) was found over Central Europe with ISORROPIA-lite predicting the higher values. With the exception of Hong Kong, the model in general overpredicted the concentrations of all three aerosol components over the East Asian region.

A comparison between ISORROPIA-lite in the metastable state and ISORROPIA II in the stable state was performed to identify potential discrepancies in the inorganic aerosol concentrations simulated by EMAC. Although differences between the two model versions are to be expected due to the different physical state of aerosols at low RH, it is of interest to examine under which conditions these differences occur so that potential users are informed of the strengths and weaknesses of using either model version depending on the application. Both modules are now available as different options in the EMAC model. The agreement between the two versions was

generally quite good for global daily mean surface concentrations of inorganic aerosols, mineral
ions and aerosol water. More specifically mineral ions, $SO_4^{2-}$ and $NH_4^+$ in TSP had the smallest
differences overall, less than 0.5 μg m$^{-3}$ even in localized extreme cases but in the vast majority
less than 0.1 μg m$^{-3}$ (or less than 5%). For coarse $NO_3^-$ aerosols the absolute differences were of
similar magnitude with the higher concentrations simulated by ISORROPIA-lite in the Middle
East being the most notable. In the case of fine $NO_3^-$ aerosols, the differences were larger (up to ~
1.75 μg m$^{-3}$ in local extremes), mainly over the West coast of South America (North of Atacama
Desert), the Tibetan Plateau and Eastern Asia regions with higher concentrations simulated by
ISORROPIA II but still within ~30%. In Europe and the US, the corresponding differences were
less than 0.25 μg m$^{-3}$. The most important difference was the higher aerosol water calculated by
ISORROPIA-lite, especially for relative humidity in the 20% to 60 % range. However, this was
less than 5 μg m$^{-3}$ or 20 % in most cases. Therefore, even though local differences are expected in
regions where the relative humidity is often in this range, on a global scale choosing a different
physical state of the aerosol at lower RH does not have such a big impact.
When the relative humidity ranged from 20 % to 60 %, differences in coarse and fine $NO_3^-$
concentrations predictions among the two versions increased. The highest discrepancies were
found in the Tibetan Plateau and the Middle East regions both of which are dominated by such RH
values during most of the year. In the first region, the combination of those RH values with mid-
range temperatures does not favor nitrate aerosol formation if the aerosol is in the metastable state
(ISORROPIA-lite). In the second region, the low RH values result in very low aerosol water
predictions for the stable state assumed by ISORROPIA II which hinder the condensation of $HNO_3$
into the aerosol phase.
Investigation of the differences in the estimated inorganic aerosol acidity between the two
versions, due to the different assumed aerosol phase states, is of great interest for potential future
use of ISORROPIA-lite in global climate simulations. ISORROPIA-lite produces slightly more
acidic coarse mode aerosols (in comparison to ISORROPIA II) but by less than 1 pH unit on
average. The most important differences were found mainly in the Middle East and Arabian
Peninsula due to the presence of high mineral cation concentrations. The stable state considered
by ISORROPIA II allows the precipitation of insoluble salts and removes anions from the aqueous
phase that would otherwise deplete the pH, while this is not the case for the metastable aerosol
state considered by ISORROPIA-lite. Furthermore, $NH_3$ is found to control the aerosol acidity of
both fine and coarse mode particles, however, it provides significantly larger buffering capacity to
the accumulation mode than to the coarse. This results in slightly more basic accumulation
particles than coarse in regions with high $NH_3$ presence from agricultural activities and low
mineral cation concentrations (e.g., Europe).
Finally concerning the computational efficiency that ISORROPIA-lite provides when used
by the EMAC global model, a speed-up of more than 3% was achieved compared to ISORROPIA
II in metastable state and nearly 5% compared to ISORROPIA II in stable state.

## Code and Data Availability

The usage of MESSy (Modular Earth Submodel System) and access to the source code is licensed to all affiliates of institutions which are members of the MESSy Consortium. Institutions can become a member of the MESSy Consortium by signing the MESSy Memorandum of Understanding. More information can be found on the MESSy Consortium Website http://www.messy-interface.org. The code developed in this study and all relevant features, including the ISORROPIA II v2.3 and ISORROPIA-lite v1.0 thermodynamic equilibrium codes as part of the MESSy system, are archived with a restricted access DOI (https://doi.org/10.5281/zenodo.8379120) and have already been incorporated into the official development branch of the EMAC modelling system and will therefore be part of all future released versions. The data produced in the study are available from the author upon request.

## Acknowledgements

This work was supported by the project FORCeS funded from the European Union's Horizon 2020 research and innovation program under grant agreement No 821205. The work described in this paper has received funding from the Initiative and Networking Fund of the Helmholtz Association through the project "Advanced Earth System Modelling Capacity (ESM)". The authors gratefully acknowledge the Earth System Modelling Project (ESM) for funding this work by providing computing time on the ESM partition of the supercomputer JUWELS (Alvarez, 2021) at the Jülich Supercomputing Centre (JSC).

## Competing Interests

HT acts as a topical editor for GMD but has no further competing interests.

## Author Contributions

AM and VAK wrote the paper with contributions from all coauthors. VAK planned the research with contributions from AKS, SNP and AN. AN and SNP provided the ISORROPIA-lite model. AM and HT performed the implementation in EMAC. AM performed the simulations and analyzed the results assisted by VAK and APT. APT provided the observations and performed the model evaluation. All the authors discussed the results and contributed to the manuscript.

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
