# Peer review of "Implementation of the ISORROPIA-lite Aerosol Thermodynamics Model into the EMAC Chemistry Climate Model (based on MESSy v2.55): Implications for Aerosol Composition and Acidity"

_Geoscientific Model Development, 2023_

## Author Comment (AC2)

**Authors' Response to Anonymous Referee's #1 Comments:**

**Summary**

*This study presents the results from the EMAC simulations when different versions of the coupled ISORROPIA thermodynamic modules are used. The study is focused on the main inorganic aerosols (i.e., $SO_4^{2-}$, $NO_3^-$, and $NH_4^+$), together with changes in the aerosol water and acidity. The authors conclude that the new version of ISORROPIA (i.e., ISORROPIA-lite) is computationally more efficient than the previous versions of the thermodynamic module (i.e., ISORROPIA II v1 and v2.3, both for stable and metastable modes) and is therefore a good replacement for 3D global simulations. The paper is well-written and well-organized, and the conclusions are useful in exploring the uncertainties of using different versions and setups of the ISORROPIA thermodynamic module in global models. However, the authors can address a few minor issues before the final publication in GMD to make the proposed parameterizations easier to understand for the reader.*

We would like to thank the reviewer for his/her thoughtful review and positive response. Below is a point-by-point response (in black) to the comments and suggestions (in blue)

**General Comments**

1. *The authors present the differences in simulating inorganic aerosols due to the various versions of the ISORROPIA thermodynamic module. Although the differences are minimal, the authors could provide some additional information on their findings. For example, the authors only state that the differences between ISORROPIA v1 and v2.3 are due to improvements in acidity calculations (Song et al. 2018), or the differences between ISORROPIA v2.3 and the lite version under the same conditions are, on average, less than 5%. Some additional sentences on the impact of these updates to the code on the simulated concentrations of inorganic aerosol components would be useful for the reader.*

We have followed the suggestion of the reviewer and added information about the differences between ISORROPIA II v1 and v2.3. We also mention that the interested reader can find additional details about these differences in Song et al. (2018). The updates in the code from ISORROPIA II v1 to v2.3 affected only a small number of the simulations in this work, in which the model failed to accurately consider the evaporation of $NH_3$. In these few instances, the pH estimated by ISORROPIA v1 was unrealistically close to neutrality. However, because this was quite rare these problems had a minimal effect on the average predicted levels of gas phase $NH_3$ and aerosol concentrations.

2. *Considering that the gas-particle partitioning of semi-volatile species such as $HNO_3$ is very sensitive to the calculated acidity levels and aerosol water concentrations, the authors could discuss more about why these differences exist in the model among the different versions of ISORROPIA, providing additional global maps for the main inorganics and focusing particularly on regions where such differences (positive or negative) are important. This, along with a slightly more detailed technical description of the advances in thermodynamic*

*calculations and the evolution of the ISORROPIA module, will help the reader understand and interpret the presented sensitivity simulations.*

Since the largest discrepancies between the various ISORROPIA versions are observed for nitrate, appropriate regions have been selected and further analyzed to investigate the source of these differences. These regions were chosen because they have high nitrate concentrations, but also because the predictions of the various ISORROPIA versions for aerosol water and acidity differ significantly over these areas. Therefore, this analysis covers the regions with high sensitivity to $HNO_3$ partitioning at least as far as ISORROPIA is concerned. A sentence clarifying this has been added towards the end of Section 4.1 of the revised manuscript. More information is also provided in Section 1 regarding the historical development of the thermodynamic calculation procedures during the evolution of ISORROPIA. Finally, Section 2.2 has been revised to include more details on the transition from ISORROPIA II to ISORROPIA-lite and the differences between the two modules.

3. *Finally, the authors present a comparison of EMAC simulations between ISORROPIA-lite and ISORROPIA II in stable mode. It is not clear why such a comparison is shown here, especially taking into account previous works of the authors. Is it because these are the standard versions available now in the EMAC model? If the "comparison is done in an attempt to quantify the effects of using the metastable case in global atmospheric simulations," as stated in the manuscript, why didn't the authors just use the metastable mode of ISORROPIA II v2.3 to show that? Wouldn't a fair comparison between the two versions require them to be in the same (metastable) mode? Further discussion is needed to support this choice since the results of the different ISORROPIA aerosol modes (i.e., stable vs. metastable) are, indeed, expected to differ.*

Indeed, it is expected that results will differ when using different ISORROPIA versions with different aerosol state assumptions. However, it is our goal to determine under which conditions and over which regions these expected differences will occur and to what extent. The reason for this is that since ISORROPIA-lite will be available alongside ISORROPIA II (in stable mode) in the new EMAC model version, it would be useful for potential users to be informed about such differences and to choose the appropriate ISORROPIA version depending on the application and the desired efficiency and/or state assumption. Further discussion has been implemented in the revised manuscript in Sections 4.1 and 5.

**Specific Comments**

4. In Sect. 4.1 (p. 14), the authors present the differences in $SO_4^{2-}$ annual mean surface concentrations between ISORROPIA-lite and ISORROPIA II (in stable mode). Does ISORROPIA II directly impact the $SO_4^{2-}$ concentrations in the model, e.g., through the formation of insoluble $CaSO_4$ and its precipitation out of the aerosol aqueous phase? Does the model also consider sulfate production in aerosol water? Does the difference in inorganics from ISORROPIA calculations impact cloud acidity in the model and, thus, the respective sulfate production? Please discuss.

ISORROPIA II has no direct impact on the predicted sulfate concentrations in the model since sulfuric acid is assumed to be practically non-volatile and to be present in the particulate phase, regardless of the state assumption used. However, differences in the predicted sulfate concentrations by EMAC in the versions using ISORROPIA-lite and ISORROPIA II in stable mode may result from indirect changes in wet deposition due to the different physical state of the aerosol. The formation of the $CaSO_4$ salt does not play a role in the predictions, since this specific salt is the only compound present in the solid state even in ISORROPIA-lite (more details can be found in Section 2.2. of the revised manuscript). Furthermore, the model does not account for sulfate production in aerosol water, but it does account for sulfate production in clouds via aqueous phase chemistry. Any differences between the two ISORROPIA versions in the inorganic aerosol ion balance (less than one pH unit) are not expected to have a significant effect on cloud acidity, which can also affect sulfate production. The higher water content in cloud droplets should smooth out any changes in aerosol acidity between the two versions, which in any case occur mostly in areas of very low RH and no cloud formation. More details about the expected differences in the predicted sulfate concentrations between the two model versions have been added to the appropriate part of Section 4.1.

5. In Sect. 4.1 (p. 14; l. 456), the authors state that the absolute differences between ISORROPIA-lite and ISORROPIA II for the fine $NO_3^-$ are greater than those of coarse mode. Although this can be explained due to the different aerosol states used for ISORROPIA among the two simulations, it would be helpful to show which version of the thermodynamic model produces results that are closer to observed values. Can such a difference in the coarse aerosols also emerge through the assumption of kinetic limitations applied in the model during condensation of $HNO_3$ in the coarse mode? Although the parameterization is well documented in the literature, a somewhat more extended discussion would be useful for the reader.

The comparison of the ISORROPIA-lite results with observations was performed in order to evaluate the new model version and to establish that it is a reliable model for the calculation of inorganic aerosol composition. The ISORROPIA II thermodynamic module has been extensively validated against observations in previous studies (De Meij et al., 2012; Pozzer et al., 2012; Karydis et al., 2016; Metzger et al., 2018). However, following the reviewer's recommendation, we have performed a statistical analysis of the comparison between observations and ISORROPIA II predictions in stable mode, which is now included in the revised supplement. Although the two versions show similar performance, it should be emphasized that better performance on certain statistical metrics should not be taken as an indication that one state assumption is more scientifically valid than the other. A corresponding discussion has been added in Section 3.3. The evaluation results of ISORROPIA-lite and ISORROPIA II in the metastable are almost identical and this is now clearly stated in the revised manuscript. Regarding the kinetic limitations during the condensation of $HNO_3$ simulated in the model, the algorithm used is the same for both ISORROPIA versions and is applied before ISORROPIA calculates the gas/particle partitioning. The algorithm assumes that the amount of $HNO_3$ that can condense in each size mode within the model time step depends on the size of the aerosol and not on its physical state. The "metastable aerosol" is expected to be larger than a "stable aerosol" due to the potentially higher amount of water it contains, but this difference is small compared to the actual size of the aerosol mode (e.g., coarse vs. accumulation mode), especially for the coarse particles. This information has been added to Section 2.2, which describes the partitioning algorithm.

6. *In Sect. 4.3 (p. 23), the authors state that the pH values are calculated based on instantaneous $H^+$ and $H_2O$ values estimated every 5 hours. Why specifically 5 hours? Does the model produce instantaneous outputs only every 5 hours by default (standard output), and that is the reason the authors use the most frequent model output? Does this, further, mean that a more frequent instantaneous output (e.g., hourly) would potentially produce more accurate pH results? Please discuss.*

The model user has the ability to control the frequency and the type (instantaneous or average values) of the output for most variables. The most commonly used, is the daily average output. However, Karydis et al. (2021) showed that a low temporal resolution output with average values can lead to a low biased calculated pH. This is due to Jensen's inequality (Jensen, 1906), which states that the convex transformation of an average value (e.g., the pH of the average $H_2O$ and $H^+$ concentrations) is less than or equal to the average applied after the convex transformation (e.g., the average of all pHs calculated based on the instantaneous $H_2O$ and $H^+$ values). For this reason, we chose to output the instantaneous values of $H_2O$ and $H^+$ instead of the average. In addition, we chose to output every 5-hour interval to always get values at different times of the day and to account for the diurnal variability of pH (i.e., not possible with 6- or 8-hour intervals). The critical choice here is the instantaneous output (instead of averages), not the time resolution. Certainly, an hourly instantaneous output would provide more accurate pH estimates, but it will also increase the size of the data produced by a factor of five. More details on this choice have been added as further discussion in Section 4.3 of the revised manuscript.

7. *Karydis et al. (2021) showed that the metastable assumption produces more acidic particles in regions with high concentrations of mineral cations, such as downwind desert areas, and low RH values. As expected, almost the same results are presented here when comparing the ISORROPIA-lite (i.e., only in the metastable mode) and ISORROPIA II (in stable mode) simulations. It is not clear, thus, the added value of such a comparison here. Can you please discuss more?*

As this study presents the first results after the implementation of ISORROPIA-lite in the EMAC global model, this comparison was performed to assess whether this version can produce credible pH estimates on a global scale. This capability of ISORROPIA II is well established in the literature (e.g. Karydis et al., 2021). Therefore, in case EMAC users decide to use ISORROPIA-lite for aerosol composition simulations, we wanted to ensure that the aerosol pH estimates are reliable and indeed similar to the estimates of ISORROPIA II using the metastable assumption. Furthermore, it is important for the user to know the differences on the estimated pH values between the two available versions of the ISORROPIA module in the new EMAC model version. More details about the inclusion of this particular comparison and further discussion about it, have been added at the beginning of Section 4.3 as well as Section 5 of the revised manuscript.

8. *It is well established that $NH_3$ is the major buffer in most regions of the world. Therefore, if all $NH_3$ emissions were turned off, the thermodynamic system would definitely give unrealistic results, and as expected, aerosol particles would be extremely acidic. Maybe doubling or cutting in half $NH_3$ emissions would make more sense to explore potential differences in the responses on the two versions. It would also be advantageous to discuss the presence of non-*

*volatile crustal species from sea salt and dust and how drastically they can change (increase) the aerosol pH in ISORROPIA-lite simulations compared to ISORROPIA v2.3 in the metastable mode. This would also give additional information on the impact of binary activity coefficient calculation between the two versions.*

We agree with the reviewer that switching off all $NH_3$ emissions results in an unrealistic thermodynamic system that would lead to very acidic aerosols. This sensitivity simulation was only performed to verify that in some regions the presence of very high $NH_3$ concentrations can lead to such an increase in the pH of the fine aerosols that it can exceed the calculated alkalinity of the coarse particles. Following the reviewer's recommendation, we performed a sensitivity simulation in which the $NH_3$ emissions were reduced by half. The results are shown in Figure 10 of the revised manuscript. In addition, while the presence of non-volatile crustal species does indeed increase aerosol pH, Karydis et al. (2021) have shown that their role in regulating aerosol acidity is small compared to the buffering provided by $NH_3$ emissions.

9. *Sect. 3.3: It would be very useful to also present the seasonal variation for the comparison of the main inorganics (where available) between observations and model predictions, not only the annual mean values. Additionally, you can present the evaluation of the other sensitivity simulations performed for this study, not only the ISORROPIA-lite. This would help the reader better understand the pros and cons of each assumption.*

We have performed a seasonal statistical analysis to compare observations and predictions of both ISORROPIA-lite and ISORROPIA II in the stable state for the three main inorganic aerosol components, since these two model versions will be available to the user in the next release of the EMAC model. Since the predictions of ISORROPIA-lite were almost identical to those of ISORROPIA II in the metastable state, the results of the latter are not shown. The discussion of Section 3.3 has been extended to include the results of the ISORROPIA II evaluation, while the tables that contain the seasonal statistical analysis can be found in the updated supplement.

10. Section 5, Page 25: It is not clear from the conclusions which version of ISORROPIA the authors propose to use for EMAC simulations. This section lacks an explanation as to why the stable mode was previously chosen for the model over the metastable mode, but now it is replaced with the metastable one. Is it only a matter of computational speed? A more detailed discussion would help.

The aim of this study is not to propose one specific version of the ISORROPIA module over the other, but rather to demonstrate that ISORROPIA-lite is equally accurate in predicting inorganic aerosols with improved computational efficiency, and to provide insight into the conditions and regions where the results of the two available versions in EMAC might differ. In previous versions of the EMAC model, the stable mode was used as the default, mainly because it was thought to represent large desert regions more realistically due to their low annual RH values (Karydis et al., 2010; Karydis et al., 2016). However, the metastable assumption is often considered more accurate for regions such as the Northeastern US (Guo et al., 2016). The choice of the default setting is now mentioned at the end of Section 2.2, and a more detailed discussion of the advantages and disadvantages of each thermodynamic state and module is given in Section 4.

11. *Sect. 2.1, l.169 & Sect. 3.1 l. 216: The emissions of crustal ions such as $Ca^{2+}$, $Mg^+$, and $K^+$, are calculated as a fraction of dust fluxes in the model. In what form these ions are emitted; totally or partially soluble/insoluble? Are these fractions directly inserted in ISORROPIA calculations? Do you also track in your model the different species upon the ISORROPIA call (e.g., $CaSO_4$)? In how many modes/sizes aerosol emissions are emitted in the model? Is ISORROPIA called for every aerosol mode/size or only for accumulation and coarse, as presented in the manuscript? If yes, how do you define here the fine aerosol acidity? Please discuss.*

   Generally, crustal ions are emitted as partially soluble/insoluble in the accumulation and coarse modes and mostly in the insoluble fraction. For this study the mineral ions $Ca^{2+}$, $Mg^+$, and $K^+$ were emitted as part of the dust flux in the insoluble fraction and in the accumulation and coarse size modes. All aerosol modes (4 soluble and 3 insoluble modes) are included in the ISORROPIA calculations as part of the system $K^+$, $Ca^{2+}$, $Mg^+$, $NH_4^+$, $Na^+$, $Cl^-$, $NO_3^-$, $SO_4^{2-}$ and $H_2O$. Insoluble particles are transferred to the soluble fraction after ISORROPIA calculations by coagulation with other soluble species, but mostly by condensation of water-soluble species (such as $HNO_3$) on their surface. EMAC tracks the concentration of all gaseous, liquid and solid species present in ISOPRROPIA, but the output is stored in the form of ions (e.g., $SO_4^{2-}$, $NO_3^-$, $NH_4^+$, etc.) for each size mode. The above information has been added in Section 2 of the revised manuscript. Aerosol acidity is only estimated for the accumulation and coarse soluble size modes. This is now clarified in Section 4.3.

**Technical Comments**

12. *Page 2, l. 50: The transition from health-related issues to the climate impacts of aerosols is very steep.*

   A connecting sentence has been added at this point in Section 1 to make the transition between health impacts and climate impacts easier for the reader.

13. *Figure 9: It would be easier for the reader to provide more details in the titles of the figures in the right column because negative pHs are acceptable values (not only for differences). A more detailed figure title can apply to all figures, especially when you show differences.*

   Titles in all figures displaying differences between any two ISORROPIA versions have been changed to be as descriptive as possible, both in the revised manuscript and in the supplement.

**References**

de Meij, A., Pozzer, A., Pringle, K. J., Tost, H., and Lelieveld, J.: EMAC model evaluation and analysis of atmospheric aerosol properties and distribution with a focus on the Mediterranean region, *Atmospheric Research*, 114-115, 38-69, https://doi.org/10.1016/j.atmosres.2012.05.014 , 2012.

Guo, H., Sullivan, A. P., Campuzano-Jost, P., Schroder, J. C., Lopez-Hilfiker, F. D., Dibb, J. E., Jimenez, J. L., Thornton, J. A., Brown, S. S., Nenes, A., and Weber, R. J.: Fine particle pH and the partitioning

of nitric acid during winter in the northeastern United States. *Journal of Geophysical Research: Atmospheres*, 121, 10, 355-310, 376, https://doi.org/10.1002/2016JD025311 , 2016.

J. L. W. V. Jensen.: Sur les fonctions convexes et les inégalités entre les valeurs moyennes. *Acta Math.* 30, 175-193, https://doi.org/10.1007/BF02418571, 1906.

Karydis, V. A., Tsimpidi, A. P., Pozzer, A., Astitha, M., and Lelieveld, J.: Effects of mineral dust on global atmospheric nitrate concentrations. *Atmospheric Chemistry and Physics*, *16*(3), 1491-1509, https://doi.org/10.5194/acp-16-1491-2016, 2016.

Karydis, V. A., Tsimpidi, A. P., Pozzer, A., and Lelieveld, J.: How alkaline compounds control atmospheric aerosol particle acidity. *Atmospheric Chemistry and Physics*, *21*(19), 14983-15001, https://doi.org/10.5194/acp-21-14983-2021, 2021.

Metzger, S., Abdelkader, M., Steil, B., and Klingmüller, K.: Aerosol water parameterization: long-term evaluation and importance for climate studies, *Atmospheric Chemistry and Physics*, 18, 16747-16774, https://doi.org/10.5194/acp-18-16747-2018, 2018.

Pozzer, A., de Meij, A., Pringle, K. J., Tost, H., Doering, U. M., van Aardenne, J., and Lelieveld, J.: Distributions and regional budgets of aerosols and their precursors simulated with the EMAC chemistry-climate model, *Atmospheric Chemistry and Physics*, 12, https://doi.org/10.5194/acp-12-961-2012, 2012.

Song, S., Gao, M., Xu, W., Shao, J., Shi, G., Wang, S. and co-authors: Fine-particle pH for Beijing winter haze as inferred from different thermodynamic equilibrium models. *Atmospheric Chemistry and Physics*, *18*(10), 7423-7438, https://doi.org/10.5194/acp-18-7423-2018, 2018.

---

## Author Comment (AC3)

**Authors' Response to Anonymous Referee's #2 Comments:**

**Summary:**

*This EMAC study investigates differences in aerosol modeling results using ISORROPIA II v1, ISORROPIA II v2.3, and ISORROPIA-lite. Notably, disparities in major aerosol components between ISORROPIA II v2.3 and ISORROPIA-lite are consistently less than 10%. Moreover, the application of ISORROPIA-lite results in a notable 5% acceleration in EMAC's computational performance. Despite ISORROPIA-lite's limitation to supersaturated aqueous (metastable) solutions, the authors endorse it as a dependable replacement for the previous thermodynamic module in EMAC. The paper's content is sufficiently detailed, and with the code now accessible through a Zenodo private repository, the manuscript could be considered for publication once all reviewer comments have been addressed. It's important to note that I concur with the specific comments made by referee 1 and won't reiterate them here.*

We thank the reviewer for the positive review of our manuscript and the helpful comments. Below is a point-by-point response to his/her comments.

**General Comments**

1. *Accuracy and Clarity: To ensure accuracy and clarity, it's essential to avoid misleading statements. While the results from ISORROPIA-lite are promising, its restriction to the metastable aerosol state renders it too limited for global atmospheric chemistry applications. This limitation could lead to errors in radiative forcing estimates, particularly in the free troposphere with low humidity. What is really needed are codes that can capture the hysteresis effect of aerosols in order to improve aerosol radiative forcing effects. Therefore, the statement that "ISORROPIA-lite can be a reliable and computationally effective replacement of the previous thermodynamic module in EMAC" should be approached with caution, pending a thorough evaluation of its suitability for global applications.*

ISORROPIA-lite should not be considered as a replacement for the ISORROPIA-II stable mode, but rather as an alternative version of the model that can be selected by the user depending on the application and the desired efficiency and/or state assumption. The aim of this study is to demonstrate that ISORROPIA-lite is equally accurate in predicting inorganic aerosol composition with improved computational efficiency and to provide insight into the conditions and regions where the results of the two versions available in EMAC might differ. However, it should be emphasized that the stable assumption should not always be considered as more accurate. During simulations, atmospheric particles are transported from one simulated cell to another by simultaneously undergoing several atmospheric processes that change their chemical composition. In many cases, they end up in computational cells with completely different RH without "carrying" their historical RH profile with them. Therefore, the choice between a stable state (e.g., following the deliquescence branch of crystallization) and a metastable state (following the efflorescence branch) should not be considered obvious. While a stable state could be considered more accurate under very low humidity conditions (e.g., over remote deserts), in regions, such as those with intermediate RH and low nitrate concentration (e.g., Northeastern US), particles are mostly in metastable state. However, the two state assumptions produce very similar results in most cases, as shown in our study. Overall, following the reviewer's comment, we have enriched our discussion in Sections 2, 4, and 5 of the revised manuscript by avoiding statements that could lead to confusion about the climatic impacts of the two model versions.

2. ***Omission of References***: *The omission of references to relevant thermodynamic codes commonly used within EMAC is a notable gap in the introduction, potentially impacting the manuscript's scientific credibility. It's crucial to acknowledge and cite widely accepted models, following established conventions in scientific publishing.*

We thank the reviewer for pointing this out. Indeed, EQSAM is the other available option besides ISORROPIA in the EMAC model for aerosol thermodynamic calculations and is now described in the introduction. We also clarify that EQSAM is still an available option in EMAC.

3. ***Consistency***: *Ensure consistency in the spelling of "ISORROPIA-lite" and other acronyms throughout the text to maintain clarity and professionalism.*

All the acronyms used in the manuscript have been thoroughly revised.

**Specific Comments:**

1. ***Discussion of Activity Coefficient***: *If tabulated activity coefficients are mentioned, it's crucial to provide a clear and comprehensive explanation or reference regarding their origin and relevance. This will ensure that readers fully understand their context.*

The use of tabulated activity coefficients (by ISORROPIA II and ISORROPIA-lite) is now explained in Section 2.2. The methodology for their calculation is briefly presented, with all relevant references cited (Kusik and HP (1978); Bromley (1973); Meissner and Peppas (1973)). Further information can be found in Fountoukis and Nenes (2007).

2. ***Temporal Analysis***: *In Table 2, where annual means of surface concentrations are discussed, it's worth noting that a 5% difference on an annual scale can translate to significantly higher variations when considering shorter timeframes, such as hourly averages, commonly used in air quality applications. To enhance the analysis, consider extending the statistical examination to at least daily values at a regional scale, focusing on selected networks. Relying solely on mean annual concentrations limits the scope of the analysis and its conclusion.*

Tables 1,2 and 7, in Sections 3.1, 3.2, and 4.1, respectively, which presented the statistical comparison between the model estimates of the different ISORROPIA versions, have been updated to include the daily averages. The box plots in Figures 7, S1 and S2, show the regional differences of the estimated daily average coarse and fine $NO_3^-$ concentrations by the different ISORROPIA versions for five specific regions. The regional analysis focuses on the differences in $NO_3^-$ concentrations since this is the aerosol component with the highest discrepancy between the different ISORROPIA versions.

3. ***Computational Speed-Up Analysis***: *The metric presented in Table 6 regarding computational speed-up should ideally encompass information about load imbalances within the system or undergo a more rigorous statistical analysis. To strengthen the analysis, consider running multiple iterations for each version to draw more robust and conclusive findings. As currently presented, the analysis is relatively weak, and its conclusions are somewhat limited.*

The statistics presented in Table 6 have been updated to include not only the results of a single simulation for each version, but a total of 18 simulations (6 for each version). The revised table 6 shows the average values of the statistical metrics used, as well as their standard deviation.

4. ***Section 4 Focus on Surface Concentrations****: Section 4 predominantly concentrates on surface concentrations, which may not offer a comprehensive evaluation of the metastable effect as intended by the authors. Consider revising the analysis in Section 4 to include an assessment of the vertical integral (burden) and, at the very least, a comparison of zonal means. The current presentation may be misleading without these additional elements.*

An assessment of the tropospheric burden of total $NO_3^-$ aerosol between the two ISORROPIA versions can be found in Section 4.1. The analysis has now been extended to include the zonal mean annual concentrations of all aerosol components and their deviation between ISORROPIA II and ISORROPIA-lite (Figures S3 and S5 in the revised supplement). We found that the deviations between the results of the two ISORROPIA versions are becoming smaller as the air masses move higher in the atmosphere, until they are practically identical at altitudes above 700hPa. The discussion in Section 4.1 has been extended accordingly.

5. ***References and Errata****: Ensure that references are not duplicated and address any missing errata. This will enhance the overall quality of the document and its accuracy.*

The reference list has been thoroughly revised.

**REFERENCES**

Bromley, L. A.: Thermodynamic properties of strong electrolytes in aqueous solutions. *AIChE journal*, *19*(2), 313-320, https://doi.org/10.1002/aic.690190216, 1973.

Fountoukis, C. and Nenes, A.: ISORROPIA II: a computationally efficient thermodynamic equilibrium model for K+–Ca 2+–Mg 2+–NH 4+–Na+–SO 4 2––NO 3––Cl––H 2 O aerosols. *Atmospheric Chemistry and Physics*, *7*(17), 4639-4659, https://doi.org/10.5194/acp-7-4639-2007, 2007

Kusik, C. and HP, M. 1978. Electrolyte Activity Coefficients in Inorganic Processing. AIChE Symp. Series, 173, 14-20, 1978.

Meissner, H. P. and Peppas, N. A.: Activity coefficients – aqueous solutions of polybasic acids and their salts, AIChE Journal,19(4), 806–809, https://doi.org/10.1002/aic.690190419, 1973.